# FLOW MATCHING FOR POSTERIOR INFERENCE WITH SIMULATOR FEEDBACK

## ABSTRACT

Flow-based models have shown great success in generative modeling, making them a promising candidate for solving inverse problems in physical sciences and allow for sampling and likelihood evaluation with much lower inference times than traditional methods. We propose to pretrain a neural network via flow matching and include control signals based on a simulator as an additional input for finetuning via a lightweight control network. Control signals can include gradients and a problem-specific cost function if the simulator is differentiable, or they can be fully learned from the simulator output. We motivate our design choices on several benchmark problems for simulation-based inference and evaluate flow matching with simulator feedback against classical MCMC methods for modeling strong gravitational lens systems, a challenging inverse problem in astronomy. We demonstrate that including simulator feedback improves the accuracy of reconstructed samples by 53%, making it competitive with traditional techniques while being up to 67x faster for inference. Upon acceptance, we will make our code publicly available.

## 1 INTRODUCTION

Acquiring posterior distributions given measurement data is of paramount scientific interest (Cranmer et al., 2020), with real-world applications ranging from particle physics (Baydin et al., 2019), over the inference of gravitational waves (Dax et al., 2021) to predictions of dynamical systems such as weather forecasting (Gneiting & Raftery, 2005). In Bayesian modeling, given an observation $x_o$ and model parameters $\theta$, we are interested in the posterior $p(\theta|x_o)$. Traditional likelihood-based methods can be expensive for high-dimensional data, when likelihood evaluations are costly or intractable and priors are difficult to represent mathematically. Simulation-based inference (Cranmer et al., 2020, SBI) addresses these challenges by including a learning-based component in the statistical inference process. In this paper, we focus on neural posterior estimation (NPE), which represents the posterior as a parametric function $q(\theta|x_o)$, which is a learnable conditional density estimator that can be trained purely by simulations $x \sim p(x|\theta)$ alone. By investing an upfront cost for training the density estimator, we can sample and compute likelihoods from $q(\theta|x_o)$ much faster than other methods, thereby amortizing the training cost over many observations. Traditionally, normalizing flows (Rezende & Mohamed, 2015; Dinh et al., 2017; Papamakarios et al., 2019) have been a popular class of density estimators used in many areas of science. To compute likelihoods and for sampling, normalizing flows transform a noise distribution to the target distribution via a bijective mapping. By conditioning the normalizing flow networks on the observation $x_o$ obtained from the simulator, they can be trained as the conditional density estimator $q(\theta|x_o)$ for the posterior. The success of diffusion models (Ho et al., 2020; Dhariwal & Nichol, 2021; Song et al., 2021) has demonstrated that the mapping between sampling and posterior distribution can be specified by a corruption process that transforms any data distribution to a normal Gaussian. Diffusion models and normalizing flows can be linked via the probability flow ODE (Song et al., 2021), which has also influenced a class of flow-based models that can be trained via flow matching (Lipman et al., 2023) on more general mappings between sampling and target distribution than considered by diffusion models. The resulting continuous-time models outperform discrete, classical normalizing flows in many areas, and training larger models is much more scalable (Wildberger et al., 2023).

Despite the widespread success of flow-based models for generative modeling and density estimation, there is no direct feedback from the simulator between the model, the observation $x_o$ and the sample

Figure 1: An overview of our proposed framework. We consider a pretrained flow network $v_\phi$ and use the predicted flow for the trajectory point $\boldsymbol{\theta}_t$ at time $t$ to estimate $\hat{\boldsymbol{\theta}}_1$. On the right, we show a gradient-based control signal with a differentiable simulator and cost function $C$ for improving $\hat{\boldsymbol{\theta}}_1$. An additional network learns to combine the predicted flow with feedback via the control signal to give a new controlled flow. By combining learning-based updates with suitable controls, we avoid local optima and obtain high-accuracy samples with low inference times.

$\boldsymbol{\theta}$ during training, which makes it very difficult to produce highly accurate samples based on learning alone.

We propose a simple strategy to reintroduce control signals using **simulators** into the flow network. We refine an existing pretrained flow-based model with a flexible control signal by aggregating the learned flow and control signals into a *controlled flow*, which requires only a minimal amount of additional parameters. To demonstrate how these refinements affect the accuracy of samples and the posterior, we consider modeling strong gravitational lens systems (Hezaveh et al., 2017; Cunha & Herdeiro, 2018; Legin et al., 2021), an inverse problem in astrophysics that is challenging and requires precise posteriors for accurate modeling of observations. In galaxy-scale strong lenses, light from a source galaxy is deflected by the gravitational potential of a galaxy between the source and observer, causing multiple images of the source to be seen. Since these images and their distortions are sensitive to the distribution of matter on small scales, this can act as a probe for different dark matter models. With upcoming and current sky surveys (Laureijs et al., 2011) expected to release large data catalogs in the near future, the number of known lenses will increase dramatically by several orders of magnitude. Traditional computational approaches require several minutes to many hours or days to model a single lens system. Therefore, there is an urgent need to reduce the compute and inference with learning-based methods. In this experiment, we demonstrate that using flow matching and our proposed control signals with feedback from a simulator, we obtain posterior distributions for lens modeling that are competitive with the posteriors obtained by MCMC-based methods but with much faster inference times.

Additionally, we evaluate different related variants of flow matching such as using problem-specific priors, self-conditioning (Chen et al., 2023) or different loss formulations in the context of SBI using several benchmark problems. We then analyze our proposed control signals for the Lotka-Volterra model, a system of coupled ordinary differential equations (ODEs) describing the population evolution of predators and prey over time. Our analysis underscores the essential role of simulator feedback for inference and that high accuracy is very challenging to achieve from scaling up datasets and model sizes alone.

To summarize, the main contributions of our work are:

- We propose a versatile strategy to improve pretrained flows with control signals based on feedback from a simulator. Control signals can be based on gradients and a cost function, if the simulator is differentiable, but they can also be learned directly from the simulator output.
- We assess different variants of flow matching in the context of SBI and demonstrate with the Lotka-Volterra model that performance gains due to simulator feedback are substantial and cannot be achieved by training on larger datasets alone.

- We demonstrate the efficacy of our proposed finetuning with control signals for inferring the parameter distributions of strong gravitational lens systems, a challenging inverse problem in astronomy that is sensitive to sample accuracy. We show that flow matching with simulator feedback is competitive with MCMC baselines and beats them significantly regarding inference time.

## 2 RELATED WORK

**Solving inverse problems under a diffusion prior**   Diffusion models have been proposed to solve linear inverse problems (Kawar et al., 2021; 2022; Chung et al., 2022; Cardoso et al., 2023), as well as general inverse problems (Holzschuh et al., 2023; Song et al., 2023; Chung et al., 2023a;b), via stochastic optimization (Graikos et al., 2022; Mardani et al., 2024) or through amortization by reinforcement learning (Black et al., 2024; Fan et al., 2023). In most of these works, the diffusion model learns the prior distribution and sampling from the posterior is achieved through a modified inference procedure, which guides samples via a conditioning. The conditioning can be based on a class label, text input (Song et al., 2021; Ho & Salimans, 2022; Saharia et al., 2022; Wu et al., 2023) or directly on a differentiable measurement operator (Chung et al., 2023a;b). In contrast to these works, we finetune a pretrained flow and learn an optimal combination of the pretrained flow and feedback from a simulator via control signals in the broader flow matching context.

**Flow matching**   Our work builds on top of prior work in flow matching (Lipman et al., 2023; Albergo et al., 2023a; Pooladian et al., 2023; Tong et al., 2023; Albergo et al., 2023b), particularly we adopt and evaluate conditional optimal transport paths (Lipman et al., 2023), test problem-specific priors and rectification of flows to produce straighter paths (Liu et al., 2023) for simulation-based inference. Guiding flows has for example been explored by Zheng et al. (2023); Nisonoff et al. (2024). We extend the existing literature by adding feedback from a simulator for scientific inverse problems.

**Simulation-based inference**   Our work directly compares to neural posterior estimation approaches for simulation-based inference (Cranmer et al., 2020; Lueckmann et al., 2021, SBI). Contrary to static architectures (Dinh et al., 2017; Kingma & Dhariwal, 2018; Papamakarios et al., 2017; Durkan et al., 2019), our approach extends the continuous-time paradigm (Chen et al., 2018; Grathwohl et al., 2019). Wildberger et al. (2023) have applied flow matching to neural posterior estimation and Sharrock et al. (2022) have used conditional diffusion models and Langevin dynamics during sampling. In contrast to previous work, we include controls signals via problem-specific simulators and cost functions during training to significantly improve the sampling quality.

**Strong lensing and parameter estimation**   Machine learning has been successfully applied to estimate parameters of lens and source models (Hezaveh et al., 2017; Levasseur et al., 2017), however, previous methods are usually restricted to point estimates, use simple variational distributions, Bayesian Neural Networks (Schuldt et al., 2021; Legin et al., 2021; Poh et al., 2022) that are not well suited to represent more complicated high-dimensional data distributions. Legin et al. (2023) predict point estimates for the lensing parameters, which are utilized by mixture density networks to model their distribution in a likelihood-free inference framework. In this paper, we combine flow matching with problem-specific simulators to obtain highly accurate samples via feedback from control signals.

## 3 FLOW MATCHING THEORY

Continuous-time flow models transform samples $\boldsymbol{\theta}$ from a sampling distribution $p_0$ to samples of a target or posterior distribution $p_1$. This mapping can be expressed via the ODE

$$d\boldsymbol{\theta}_t = v_\phi(t, \boldsymbol{\theta}_t)dt, \tag{1}$$

where $v_\phi(t, \boldsymbol{\theta}_t)$ represents a neural network with parameters $\phi$. Early works (Chen et al., 2018; Grathwohl et al., 2019) optimize $v_\phi(t, \boldsymbol{\theta})$ using maximum likelihood training, which is computationally demanding and difficult to scale to larger networks. Instead, in flow matching the network $v_\phi(t, \boldsymbol{\theta})$ is trained by regressing a vector field $u(t, \boldsymbol{\theta})$ that generates probability paths that map from $p_0$ to $p_1$.

**Generating probability paths**   We say that a smooth[1] vector field $u : [0, 1] \times \mathbb{R}^d \to \mathbb{R}^d$, called *velocity*, generates the probability paths $p_t$, if it satisfies the continuity equation $\frac{\partial p}{\partial t} = -\nabla \cdot (p_t u_t)$

---

[1]the vector field $u$ is locally Lipschitz in $\boldsymbol{\theta}$ and Bochner integrable in $t$

when viewed as a function $p : [0, 1] \times \mathbb{R}^d \to \mathbb{R}$. Informally, this means that we can sample from the distribution $p_t$ by sampling $\boldsymbol{\theta}_0 \sim p_0$ and then solving the ODE $d\boldsymbol{\theta} = u(t, \boldsymbol{\theta})dt$ with initial condition $\boldsymbol{\theta}_0$. In the following, we will denote $u(t, \boldsymbol{\theta})$ by $u_t(\boldsymbol{\theta})$. To regress the velocity field, we define the **flow matching** objective

$$\mathcal{L}_{\mathrm{FM}}(\theta) := \mathbb{E}_{t \sim \mathcal{U}(0,1), \boldsymbol{\theta} \sim p_t(\boldsymbol{\theta})} \left|\left| v_\theta(t, \boldsymbol{\theta}) - u_t(\boldsymbol{\theta}) \right|\right|^2 . \tag{2}$$

In order to compute this loss, we need to sample from the probability distribution $p_t(\boldsymbol{\theta})$ and we need to know the velocity $u_t(\boldsymbol{\theta})$. However, in general $u_t(\boldsymbol{\theta})$ is not accessible.

**Conditioning variable**   To solve this problem, we apply a trick by introducing a latent variable $\boldsymbol{z}$ distributed according to $q(\boldsymbol{z})$ and define the conditional likelihoods $p_t(\boldsymbol{\theta}|\boldsymbol{z})$ that depend on the latent variable so that $p_t(\boldsymbol{\theta}) = \int p_t(\boldsymbol{\theta}|\boldsymbol{z})q(\boldsymbol{z})d\boldsymbol{z}$. Interestingly, if the conditional likelihoods are generated by the velocities $u_t(\boldsymbol{\theta}|\boldsymbol{z})$, then the velocity $u_t(\boldsymbol{\theta})$ can be written in terms of $u_t(\boldsymbol{\theta}|\boldsymbol{z})$ and $p_t(\boldsymbol{\theta}|\boldsymbol{z})$ with $u_t(\boldsymbol{\theta}) := \mathbb{E}_{q(\boldsymbol{z})}[u_t(\boldsymbol{\theta}|\boldsymbol{z})p_t(\boldsymbol{\theta}|\boldsymbol{z})/p_t(\boldsymbol{\theta})]$. We can choose paths $p_t(\boldsymbol{\theta}|\boldsymbol{z})$ that are easy to sample from and for which we know the generating velocities $u_t(\boldsymbol{\theta}|\boldsymbol{z})$. Next, we define the **conditional flow matching** loss

$$\mathcal{L}_{\mathrm{CFM}}(\phi) := \mathbb{E}_{t, q(\boldsymbol{z}), p_t(\boldsymbol{\theta}|\boldsymbol{z})} \left|\left| v_\phi(t, \boldsymbol{\theta}) - u_t(\boldsymbol{\theta}|\boldsymbol{z}) \right|\right|^2 . \tag{3}$$

In contrast to the flow matching loss eq. 2, this loss is tractable and can be used for optimization. Now, one can show (Tong et al., 2023) that if $p_t(\boldsymbol{\theta}) > 0$ for all $\boldsymbol{\theta} \in \mathbb{R}^d$, then

$$\nabla_\phi \mathcal{L}_{\mathrm{FM}}(\phi) = \nabla_\phi \mathcal{L}_{\mathrm{CFM}}(\phi). \tag{4}$$

This means that we can train $v_\theta(\boldsymbol{\theta}, t)$ to regress $u_t(\boldsymbol{\theta})$ generating the mapping between $p_0$ and $p_1$ by optimizing the conditional flow matching loss eq. 3.

**Couplings**   The above framework allows for many degrees of freedom when specifying the mapping from $p_0$ to $p_1$ via the conditioning variable $\boldsymbol{z}$ and the conditional likelihoods $p_t$. One particularly intuitive and simple choice is to consider the coupling $q(\boldsymbol{z}) = p_1(\boldsymbol{\theta})$, i.e. the conditioning variable $\boldsymbol{z}$ is identified with the endpoint $\boldsymbol{\theta}_1$ (Lipman et al., 2023), together with conditional probability and generating velocity

$$p_t(\boldsymbol{\theta}|\boldsymbol{\theta}_1) = \mathcal{N}(\boldsymbol{\theta}| t\boldsymbol{\theta}_1, (1 - (1 - \sigma_{\min})t)I) \quad \text{and} \quad u_t(\boldsymbol{\theta}|\boldsymbol{\theta}_1) = \frac{\boldsymbol{\theta}_1 - (1 - \sigma_{\min})\boldsymbol{\theta}}{1 - (1 - \sigma_{\min})t}, \tag{5}$$

where $\sigma_{\min} > 0$. Conditioned on $\boldsymbol{\theta}_1$, this coupling transports a point $\boldsymbol{\theta}_0 \sim \mathcal{N}(0, I)$ from the sampling distribution to the posterior distribution on the linear trajectory $t\boldsymbol{\theta}_1$ ending in $\boldsymbol{\theta}_1$ but decreasing the standard deviation from 1 to a smoothing constant $\sigma_{\min}$. In this case, the transport path coincides with the optimal transport between two Gaussian distributions.

## 4   CONTROLS FOR IMPROVED ACCURACY

While flow-based models $v_\phi(t, \boldsymbol{\theta})$ gradually transform samples from $p_0$ to $p_1$ in many steps during inference via solving the ODE eq. 1, there is no direct feedback loop between the underlying simulator, the current point on the trajectory $\boldsymbol{\theta}_t$, and the observation $\boldsymbol{x}_o$. A central goal of our work is to reintroduce this feedback loop into inference and training by incorporating a control signal.

**Conditioning of flows**   Flows $v_\phi(t, \boldsymbol{\theta})$ can be conditioned on an observation $\boldsymbol{x}_o$ through an additional input $v_\phi(t, \boldsymbol{\theta}, \boldsymbol{x}_o)$, therefore modeling the conditional densities $p_t(\boldsymbol{\theta}|\boldsymbol{x}_o)$ (Song et al., 2021). Models can be trained for both conditional and unconditional generation. This is achieved, for example, in classifier free-guidance (Ho & Salimans, 2022), by randomly dropping the conditioning and setting it to 0 during training.

A critical shortcoming here is that the conditioning $\boldsymbol{x}_o$ is static, whereas we propose to have a dynamic control mechanism that depends on the trajectory $\boldsymbol{\theta}_t$, the observation, and an underlying control signal. The latter should relate $\boldsymbol{\theta}_t$ and observation using a physics-based model represented through a cost function $C$. As the accuracy of neural networks is inherently limited by the finite size of their weights, and smaller networks are attractive from a computational perspective, physics-based control has the potential to yield high accuracy with lean and efficient neural network models.

**1-step prediction** An additional issue is that the current trajectory $\boldsymbol{\theta}_t$ might not be close to a good estimate of a posterior sample $\boldsymbol{\theta}_1$, especially at the beginning of inference, where $\boldsymbol{\theta}_0$ is drawn from the sampling distribution. This issue is alleviated by applying the cost function $C$ to the current estimate $\boldsymbol{\theta}_t$, we extrapolate $\boldsymbol{\theta}_t$ forward in time to obtain an estimated $\hat{\boldsymbol{\theta}}_1$

$$\hat{\boldsymbol{\theta}}_1 = \boldsymbol{\theta}_t + (1-t)v_\phi(t, \boldsymbol{\theta}_t, \boldsymbol{x}_o). \tag{6}$$

This estimate is exact, if the trained model perfectly fits the conditional optimal transport paths.

**Comparison with likelihood-guidance** The 1-step prediction is conceptually related to diffusion sampling using likelihood-guidance (Chung et al., 2022; Wu et al., 2023). For inference in diffusion models, sampling is based on the conditional score $\nabla_{\boldsymbol{\theta}_t} \log p(\boldsymbol{\theta}_t|\boldsymbol{x}_o)$, which can be decomposed into

$$\nabla_{\boldsymbol{\theta}_t} \log p(\boldsymbol{\theta}_t|\boldsymbol{x}_o) = \nabla_{\boldsymbol{\theta}_t} \log p(\boldsymbol{\theta}_t) + \nabla_{\boldsymbol{\theta}_t} \log p(\boldsymbol{x}_o|\boldsymbol{\theta}_t). \tag{7}$$

The first expression can be estimated using a pretrained diffusion model, whereas the latter is usually intractable, but can be approximated using $p(\boldsymbol{x}_o|\boldsymbol{\theta}_t) \approx p_{\boldsymbol{x}_o|\boldsymbol{\theta}_0}(\boldsymbol{x}_o|\hat{\boldsymbol{\theta}}(\boldsymbol{\theta}_t))$, where the denoising estimate $\hat{\boldsymbol{\theta}}(\boldsymbol{\theta}_t) := \mathbb{E}_q[\boldsymbol{\theta}_0|\boldsymbol{\theta}_t]$ is usually obtained via Tweedie's formula $(\mathbb{E}_q[\boldsymbol{\theta}_0|\boldsymbol{\theta}_t] - \boldsymbol{\theta}_t)/t\sigma^2$. In practice, the estimate $\hat{\boldsymbol{\theta}}(\boldsymbol{\theta}_t)$ is very poor when $\boldsymbol{\theta}_t$ is still noisy, impeding the inference in the early stages. On the contrary, flows based on linear conditional transportation paths have empirically been shown to have trajectories with less curvature (Lipman et al., 2023) compared to, for example, diffusion models, thus enabling inference in fewer steps and providing better estimates for $\hat{\boldsymbol{\theta}}_1$.

**Controlled flow $v_\phi^C$** We pretrain the flow network $v_\phi(t, \boldsymbol{\theta}, \boldsymbol{x}_o)$ without any control signals to make sure that we can realize the best achievable performance possible based on learning alone. Then, in a second training phase, we introduce the control network $v_\phi^C(t, \boldsymbol{v}, \boldsymbol{c})$ with pretrained flow $\boldsymbol{v}$ and control signal $\boldsymbol{c}$ as input. The control network is much smaller in size than the flow network, making up ca. $10\%$ of the weights $\phi$ in our large-scale experiments. We freeze the network weights of $v_\phi$ and train with the conditional flow matching loss eq. 3 for a small number of additional steps. This reduces training time and compute since we do not need to backpropagate gradients through $v_\phi(t, \boldsymbol{\theta}, \boldsymbol{x}_o)$. We did not observe that freezing the weights of $v_\phi$ affects the performance negatively. We include algorithms for training in appendix A.

### 4.1 Types of control signals

Aiming for high inference accuracy, we extend self-conditioning via physics-based control signals to include an additional feedback loop between the model output and an underlying physics-based prior. We distinguish between two types of control signals.

**Gradient-based control signal** In the first case, there is a differentiable cost function $C$ and a deterministic differentiable simulator $S$ as shown in fig. 2a. Given an observation $\boldsymbol{x}_o$ and the estimated prediction $\hat{\boldsymbol{\theta}}_1$, the control signal relates to how well $\hat{\boldsymbol{\theta}}_1$ explains $\boldsymbol{x}_o$ via some cost function $C$. The cost function can also depend directly on or be equal to the likelihood $p(\boldsymbol{x}_o|\hat{\boldsymbol{\theta}}_1)$. For a differentiable cost function $C$, we define the control signal via

$$\boldsymbol{c}(\hat{\boldsymbol{\theta}}_1, \boldsymbol{x}_o) := [C(S(\hat{\boldsymbol{\theta}}_1), \boldsymbol{x}_o); \nabla_{\hat{\boldsymbol{\theta}}_1} C(S(\hat{\boldsymbol{\theta}}_1), \boldsymbol{x}_o)]. \tag{8}$$

We can use any control that depends on $\hat{\boldsymbol{\theta}}_1$ and $\boldsymbol{x}_o$ and is informative for the given task.

**Learning-based control signal** In the second case, the simulator is non-differentiable. To combine the simulator output with the observation $\boldsymbol{x}_o$, we introduce a learnable encoder model *Enc* with parameters $\phi_E$. The output of the encoder is small and of size $O(\dim(\boldsymbol{\theta}))$. The control signal is then defined as

$$\boldsymbol{c}(\hat{\boldsymbol{\theta}}_1, \boldsymbol{x}_o) := Enc(S(\hat{\boldsymbol{\theta}}_1), \boldsymbol{x}_o). \tag{9}$$

The gradient backpropagation is stopped at the simulator output, see fig. 2b.

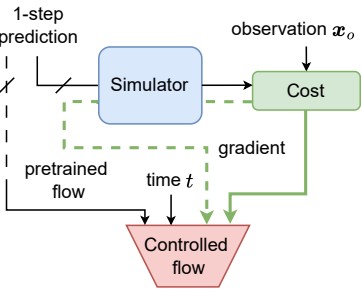

(a) Gradient-based control signal

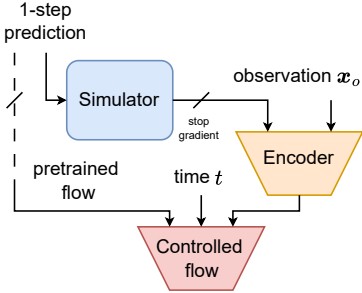

(b) Learning-based control signal

Figure 2: Control signals with simulator feedback.

## 4.2 Additional Considerations for Simulator Feedback

**Stochastic simulators**   Many Bayesian inference problems have a stochastic simulator. For simplicity, we assume that all stochasticity within such a simulator can be controlled via a variable $z \sim \mathcal{N}(0, I)$, which is an additional input. Motivated by the equivalence of exchanging expectation and gradient

$$\nabla_{\hat{\boldsymbol{\theta}}_1} \mathbb{E}_{z \sim \mathcal{N}(0,1)}[C(S_z(\hat{\boldsymbol{\theta}}_1), \boldsymbol{x}_o)] = \mathbb{E}_{z \sim \mathcal{N}(0,1)}[\nabla_{\hat{\boldsymbol{\theta}}_1} C(S_z(\hat{\boldsymbol{\theta}}_1), \boldsymbol{x}_o)], \qquad (10)$$

when calling the simulator, we draw a random realization of $z$. During training, we randomly draw $z$ for each sample and step while during inference we keep the value of $z$ fixed for each trajectory.

**Time-dependence**   If the estimate $\hat{\boldsymbol{\theta}}_1$ is bad and the corresponding cost $C(\hat{\boldsymbol{\theta}}_1, \boldsymbol{x}_o)$ is high, gradients and control signals can become unreliable. In appendix B, we empirically find that the estimates $\hat{\boldsymbol{\theta}}_1$ become more reliable for $t \geq 0.8$. Therefore, we only train the control network $v_\phi^C$ in this range, which allows for focusing on control signals containing the most useful information. For $t < 0.8$, we directly output the pretrained flow $v_\phi(t, \boldsymbol{\theta}, \boldsymbol{x}_o)$.

**Theoretical correctness**   Contrary to likelihood-based guidance, which uses an approximation for $\nabla_{\boldsymbol{\theta}_t} \log p(\boldsymbol{x}_o|\boldsymbol{\theta}_t)$ as a guidance term during inference, the approximation $\hat{\boldsymbol{\theta}}_1$ only influences the control signal, which is an input to the controlled flow network $v_\phi^C$. In the case of a deterministic simulator, this makes the control signal a function of $\boldsymbol{\theta}_t$. The controlled flow network is trained with the same loss as vanilla flow matching (Lipman et al., 2023). Therefore all theoretical properties remain preserved.

## 5 Simulation-based Inference

This section is organized as follows. First, in section 5.1, we introduce a set of SBI benchmark tasks and provide a comparison of popular neural posterior estimation (NPE) methods against a baseline of flow matching without simulator feedback. This comparison uses a similar training setup for all models and tasks. Then, in section 5.2, we focus on an optimal task-specific network with training hyperparameters based on an extensive grid search. We evaluate different variants of flow matching that are related to simulator feedback on the SBI tasks to push the performance as far as possible. In section 5.3, we pick the most challenging SBI task and improve it further by introducing simulator feedback via gradient-based and learned control signals. We carefully analyze the cost-accuracy trade-off for using simulators and show that improvements from simulator feedback cannot be replicated by increasing the training dataset size alone.

### 5.1 Tasks and baselines

We consider the SBI tasks Lotka Volterra **LV**, a coupled ODE for the population dynamics of interacting species, **SIR**, an epidemiological model for the spread of diseases, **SLCP** and Two Moons (**TM**), two synthetic tasks having complicated multimodal posteriors. All tasks are part of the benchmark collection from Lueckmann et al. (2021). For each problem, the posterior distribution for a set of 10 observations is known, which allows for directly comparing it with the posterior predicted by the trained model. This is

Table 1: C2ST comparison with identical training setups and comparable number of network weights (ca. 300K).

| Method | LV | SLCP | SIR | TM |
|---|---|---|---|---|
| CNF | 0.99 | 0.80 | 0.99 | 0.60 |
| NSF | 0.99 | - | **0.75** | **0.54** |
| FFJORD | 0.95 | 0.82 | 0.78 | 0.59 |
| *Flow-Mat.* | **0.93** | **0.79** | 0.79 | 0.58 |

measured using the C2ST score (Lopez-Paz & Oquab, 2017), which trains a classifier to discriminate between samples from the true posterior and samples generated from the learned model. If the classifier cannot discriminate between two sets of samples, its test accuracy will be 0.5, whereas it increases when they become more dissimilar.

We include the following baseline methods for NPE: Continuous normalizing flows (Chen et al., 2018, CNF), Neural Spline Flows (Durkan et al., 2019, NSF), and FFJORD (Grathwohl et al., 2019). Since we propose to include feedback from simulators, here we focus on the largest benchmark budget of $10^5$ simulator calls for generating the training dataset. Table 1 highlights that flow matching yields a highly competitive performance in this setting. For details on the training setup, see appendix B.

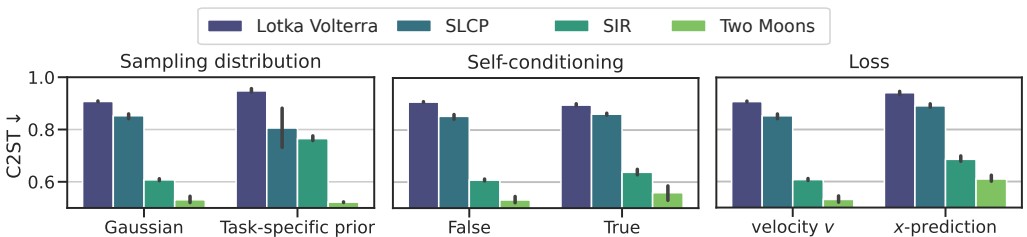

Figure 3: Evaluation of SBI tasks using different variants of flow matching training. Lower C2ST scores are better.

Flow matching has also been evaluated for the SBI benchmark tasks by Wildberger et al. (2023), who performed an extensive hyperparamter search for each task to find optimal hyperparameters. In the following, we focus on flow matching, and hence use the corresponding sets of optimal hyperparameters for each task.

## 5.2 TRAINING VARIANTS

There are several variants of training diffusion models that can be related to simulation feedback and which we consider promising in the context of SBI. Before we go on to evaluate the simulator feedback in section 5.3, we test if we can improve the performance using any of them. In particular, we assess the following modifications:

- **Self-conditioning**: conditioning a model on something that depends on its own output can be seen as a form of self-conditioning. We evaluate an adapted version of self-conditioning (Chen et al., 2023). Instead of providing $\boldsymbol{\theta}_t$ to the flow network, the input is comprised of the concatenated vector $[\boldsymbol{\theta}_t; \mathrm{Dropout}(\hat{\boldsymbol{\theta}}_1)]$, where $\hat{\boldsymbol{\theta}}_1$ is the 1-step prediction eq. 6. For computing $\hat{\boldsymbol{\theta}}_1$, we require one network evaluation with the input $[\boldsymbol{\theta}_t; 0]$ and stop the gradient backpropagation at $\hat{\boldsymbol{\theta}}_1$. This method is similar to our simulator feedback, as it introduces a feedback loop that conditions the model on its own output, but without any simulator.

- **Task-specific priors**: it is also possible to couple two non-Gaussian distributions by defining the coupling as $q(\boldsymbol{z}) = p_0(\boldsymbol{\theta}_0)p_1(\boldsymbol{\theta}_1)$ and setting the conditional probabilities to the linear paths defined by $p_t(\boldsymbol{\theta}|(\boldsymbol{\theta}_0, \boldsymbol{\theta}_1)) = \mathcal{N}(\boldsymbol{\theta}|t\boldsymbol{\theta}_1 + (1-t)\boldsymbol{\theta}_0, \sigma I)$ and $u_t(\boldsymbol{\theta}|(\boldsymbol{\theta}_0, \boldsymbol{\theta}_1)) = \boldsymbol{\theta}_1 - \boldsymbol{\theta}_0$ with bandwidth $\sigma > 0$. We can choose $p_0$ as the prior distribution $p(\boldsymbol{\theta})$ which we know in the SBI setting. Obtaining information in the form of an observation changes our knowledge about $\theta$ from the prior distribution to the posterior, therefore resembling a transformation similar to the noise to data transformation in diffusion models. This also suggests that the prior distribution can be closer to the posterior than a noise distribution.

- $x$-**prediction**: the reliability of the control signal depends directly on the 1-step estimate $\hat{\boldsymbol{\theta}}$. Instead of regressing the flow $u_t(\theta)$, we can directly predict the denoised estimate $\hat{\boldsymbol{\theta}}$ and obtain the velocity by rearranging eq. 6, giving $v_\phi(t, \boldsymbol{\theta}_t, \boldsymbol{x}_o) = \hat{\boldsymbol{\theta}}_1/(1-t)$. We additionally weight the $x$-prediction loss with a time-dependent weighting $w_t := 1/(1-t)$ to account for the scaling in eq. 6. The $x$-prediction potentially produces better estimates for $\hat{\boldsymbol{\theta}}$, thus allowing for obtaining more reliable feedback from control signals when $t < 0.8$.

**Evaluation** Figure 3 shows an evaluation of the different variants against vanilla flow matching (Gaussian sampling distribution, no self-conditioning and velocity prediction). Using task-specific priors produces outliers with better C2ST scores for SLCP but is consistently worse for LV and SIR. We conclude that normal Gaussian distributions are more suited as sampling distributions for most low-dimensional problems. Introducing self-conditioning does not show any improvements, so feedback loops without a simulator alone are not sufficient for better performance in this situation. Finally, the $x$-prediction loss consistently performs worse than the velocity prediction. Therefore, a potential improvement in the 1-step estimate is outweighed by a corresponding deterioration of the posterior correctness as indicated by the C2ST score.

## 5.3 Simulator feedback: Gradient-based and Learned

In this section, we focus on the Lotka-Volterra (LV) task for a more detailed analysis. It has the highest difficulty as seen by the C2ST score, and we use it to test the different types of feedback. We reimplement the LV simulator in JAX (Bradbury et al., 2018) to support differentiability and evaluate the gradient-based control signal as well as the learning-based control signal, using a small multilayer perceptron (MLP). In addition, to make sure that observed improvements are not due to the increased number of network parameters and finetuning with the control network, we also evaluate a variant where we finetune with the control network but set all simulator-dependent inputs to the control network to 0 (Zero Controls). We show an evaluation with C2ST in fig. 4. For both the learning and gradient-based control signals we see clear improvements with the gradient-based signal clearly ahead. The zero control signal improves only slightly, showing that the improvement can be directly attributed to the simulator.

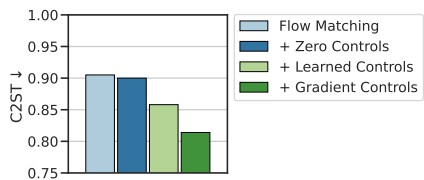

Figure 4: Evaluation of simulator feedback for **LV**.

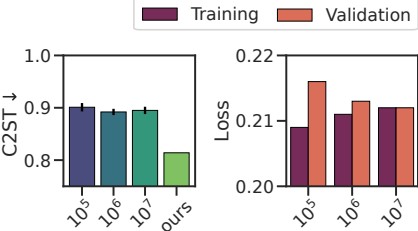

Figure 5: Different simulator call budgets (training set sizes $10^5$, $10^6$, $10^7$) compared with finetuning using simulator feedback (ours, ca. $9 \times 10^6$ simulator calls in total).

While control signals are most useful for more high-dimensional problems with less sparse and noisy observations, this experiment demonstrates that they can also be used in low-dimensional settings. Moreover, while differentiable simulators can provide better control signals, feedback from non-differentiable simulators likewise shows clear improvements.

## 5.4 Computational efficiency

A critical issue in SBI is that calls to the simulator are potentially expensive. This imposes the question of whether compute time is better spent on extending the training dataset or training with feedback from the simulator. We empirically verify that the latter is more efficient for the LV task in this setup by comparing our method to models with an increased training dataset from a larger simulator budget. Specifically, we train with dataset sizes of $10^6$ and $10^7$. Training the gradient-based control signal took ca. $9 \times 10^6$ simulator calls. See fig. 5 for the evaluation. There is no improvement in the C2ST for models trained without simulator feedback beyond $10^5$ data points, and the final train/validation loss for the $10^7$ model indicates that there is no more overfitting. Nonetheless, the model trained with controls clearly outperforms the model trained with more data, indicating that the directed feedback of the simulator cannot be replaced by increased amounts of training data.

## 6 Strong Gravitational Lensing

We present our results for modeling strong gravitational lens systems, a challenging and highly relevant non-linear problem in astronomy. Strong gravitational lensing is a physical phenomenon whereby the light rays by a distant object, such as a galaxy, are deflected by an intervening massive object, such as another galaxy or a galaxy cluster. As a result, one observes multiple distorted images of the background source. We aim to recover both the lens and source light distribution as well as the lens mass density distribution with realistic simulated observations for which we know the ground truths. We evaluate flow matching as an NPE method with gradient-based control signals from a differentiable simulator with two MCMC methods.

**Lens modeling** The *lens equation* relates coordinates on the source plane $\beta$ and the observed image plane $\Theta$ via the deflection angle $\alpha$ induced by the mass profile or gravitational potential of the lens galaxy. We use a Singular Isothermal Ellipsoid (SIE) to describe this lens mass and Sérsic profiles for both the source light and light emitted from the lens galaxy (full details are provided in appendix C). There are 9 parameters for the lens mass and 7 parameters for each Sérsic profile, giving 23 parameters in total. The likelihood is measured by the $\chi^2$-statistic, which is the modeled image plane $\Theta$ minus the observation $\boldsymbol{x}_o$ divided by the noise. To solve the lensing equation, we make use of

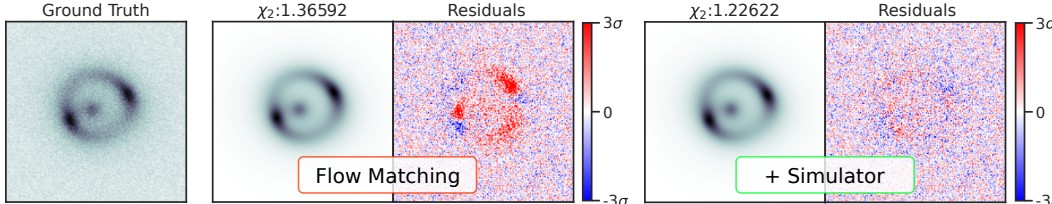

Figure 6: Reconstruction of observation. Flow matching is purely learning-based and shows noticeable residuals in the reconstruction. Including simulator feedback removes remaining residuals.

the publicly available raytracing code by (Galan et al., 2022). We want to stress that even small perturbations of the model parameters can cause the $\chi_2$ to increase significantly; see fig. 14 in the appendix.

**Datasets and pretraining**  Several instrument-specific measurement effects are included when simulating the observations. We include background and Poisson noise and smoothing by a point-spread function (PSF). The pixel size corresponds to $0.04$ arc seconds. These directly affect the posterior, as more noise and a stronger PSF will widen the posterior distribution. We generate $250\,000$ data samples for training and $25\,000$ for validation. The flow network $v_\phi$ consists of a convolutional feature extraction neural network represented by a shallow CNN whose output is fed into a dense feed-forward neural network with residual blocks. Full details are in appendix C.

**Finetuning with control signals**  The control network $v_\phi^C$ is represented by another dense feed-forward network, which accounts for $11\%$ of all parameters in the combined model. The control signals are obtained from simulating an observation based on the predicted estimate $\hat{\boldsymbol{\theta}}_1$ via ray-tracing (Galan et al., 2022) based on the parametric models, calculating the $\chi_2$-statistic and computing gradients with respect to the estimate $\hat{\boldsymbol{\theta}}_1$. The $\chi_2$-statistic itself is also part of the control signal.

**Reference posteriors**  As reference posteriors, we include Hamiltonian Monte Carlo (HMC) with No-U-Turn sampler (Hoffman et al., 2014, NUTS) and Affine-Invariant Ensemble Sampling (Goodman & Weare, 2010, AIES), which are both two popular MCMC-methods in astronomy. We adopt implementations of both methods using numpyro (Phan et al., 2019; Bingham et al., 2019). Additionally, we compare to diffusion posterior sampling (Chung et al., 2023b, DPS), loss-guided diffusion (Song et al., 2023, LGD-MC) and twisted diffusion sampler (Wu et al., 2023, TDS). Details on all baseline methods can be found in appendix C. We use Euler integration for both flow matching variants.

### 6.1 EVALUATION AND DISCUSSION

$\chi_2$**-statistic**  We show an evaluation of all methods in table 2. The average $\chi_2$ is computed over 1000 randomly chosen validation systems, where for each, we draw 1000 samples from the posterior. If we compute the $\chi_2$ for the ground truth parameters, we obtain a value of $1.17$ due to the noise in the observation. Since we cannot overfit to noise with the parametric models, this represents a lower bound for $\chi_2$ in this experiment. Including the physics-based control improves the $\chi_2$ from $1.83$ to $1.48$, representing an improvement of $53\%$ relative to the best modeling. The improved $\chi_2$ is even better than the best baseline method, AIES.

Table 2: Evaluation with respect to average $\chi_2$ and inference time for the posterior distribution.

| Method | Avg. $\chi_2 \downarrow$ | Modeling Time $\downarrow$ |
|---|---|---|
| NUTS | 1.83 | $\sim$ 56x (564s) |
| AIES | 1.74 | $\sim$ 67x (672s) |
| DPS | 9.98 | $\sim$ 42x (427s) |
| LGD-MC(5) | 21.62 | $\sim$ 160x (1600s) |
| TDS (k=100) | 20.94 | $\sim$ 21x (210s) |
| *Flow-Mat.* | 1.83 | 1x (**10s**) |
| + Simulator | **1.48** | $\sim$ 2x (19s) |

**Modeling time**  We define the modeling time as the average compute time required to produce 1000 credible posterior samples. Both HMC and AIES require significant warmup times before producing the first samples from the posterior, which we include in the table. However, after warmup, it is relatively cheap to obtain new samples. On the other hand, flow matching does not require

any warmup time and the modeling time increases linearly with the number of posterior samples. All methods were implemented in JAX (Bradbury et al., 2018) and used the same hardware. The measurements in table 2 show that DPS is faster than the classic baselines, but yields a very sub-optimal performance in terms of its distribution. The performance numbers also highlight that our method yields an accuracy that surpasses AIES, while being more than 30x faster.

This evaluation demonstrates that flow matching-based methods are highly competitive even in small to moderate-sized problems where established MCMC methods in terms of accuracy exist, clearly beating them in terms of inference time. Flow matching with our proposed control signals is especially interesting because it is not affected as much by the curse of dimensionality as traditional inference methods and allows for having non-trivial learnable high-dimensional priors. However, before these methods are widely trusted, they need to demonstrate their competitiveness with classical methods. Our results show that this is indeed the case, which opens up exciting avenues for applying and developing approaches targeting similar and adjacent inverse problems in science.

**Simulation-based calibration**  Acquiring truthful posterior distributions for Bayesian inference problems is difficult, which makes it hard to robustly evaluate whether the predicted posterior distribution is correct. We use simulation-based calibration (Talts et al., 2018, SBC) as an additional evaluation tool. The data-averaged posterior obtained from averaging the posterior distribution over many problem instances has to be equal to the prior. This can be tested by considering a one-dimensional function $f : \boldsymbol{\theta} \mapsto \mathbb{R}$ and $L$ samples $\boldsymbol{\theta}^1, ..., \boldsymbol{\theta}^L$ drawn from an inference method. If $\boldsymbol{\theta}^*$ are the ground truth parameters, then the rank statistic $\sum_{l=1}^{L} \mathbf{1}_{f(\boldsymbol{\theta}^L) < f(\boldsymbol{\theta}^*)}$ has to be uniformly distributed over the integers $[0, L]$. If the distribution of the rank statistic is plotted as a histogram, systematic problems in the inference method can be identified visually, see fig. 7. We set $L = 1000$ and plot the histograms for all $n = 1000$ test problems and visualize the parameter $x_{\text{center}}$, which defines the position of the source in $x$-direction. The posteriors without simulator feedback are biased, as can be

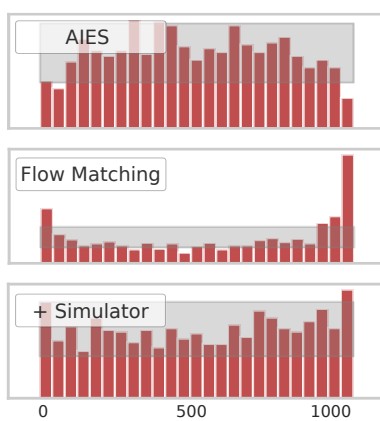

Figure 7: SBC for $x_{\text{center}}$ of the source galaxy.

seen in the deviation from uniformity in the plots. Including simulator feedback improves the distribution of the rank statistic. For an extended analysis, see appendix C.4.

## 6.2 LIMITATIONS

While introducing additional control signals increases the quality of produced samples, it comes at the cost of slower inference and training times depending on the speed of the simulator. In general, using non-differentiable control signals is possible but removes the possibility of computing likelihoods via the instantaneous change of variables formula (Chen et al., 2018). Compared to MCMC approaches, inference with flow-based models requires a substantial upfront cost for training that needs to be amortized across many problems. Additionally, priors are encoded in the learned flow networks, so changing them would require retraining models with adjusted data sets.

## 7 CONCLUSION

We presented a method for improving flow-based models with simulator feedback using control signals. This allows us to refine an existing flow with only a few additional weights and little training time. We thereby efficiently bridge the gap between purely learning-based methods for simulation-based inference and optimization with hand-crafted cost functions within the framework of flow matching. This improvement is critical for scientific applications where high accuracy and trustworthiness in the methods are required. Purely learning-based methods face significant difficulties in producing very accurate samples, as there is usually no feedback during inference of how good samples are. In this paper, we demonstrated that we do not need large network sizes or tremendous amounts of data to train accurate models that are competitive with established MCMC methods if we include suitable control signals from simulators. We believe this work makes an important step towards making posterior inference in science more accurate, understandable, and reliable.

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

# APPENDIX

## A ALGORITHMS

We include algorithms for training using flow matching and control signals, see algorithm 1. For flow matching with self-conditioning, see algorithm 2.

---

**Algorithm 1** FM with Control Signals

**Input:** Training distribution $q_1$, pretrained network $v_\phi$, control network $v_\phi^C$, $\sigma_{\min}$
**while** Training **do**
$\quad (\boldsymbol{\theta}_1, \boldsymbol{x}_o) \sim q_1; \; z \leftarrow \mathcal{N}(0, I)$
$\quad \boldsymbol{\theta} \leftarrow t\boldsymbol{\theta}_1 + (1-t)\boldsymbol{z}$
$\quad \boldsymbol{v} \leftarrow \text{stopgrad}(v_\phi(t, \boldsymbol{\theta}, \boldsymbol{x}_o))$
$\quad \hat{\boldsymbol{\theta}}_1 \leftarrow \boldsymbol{\theta} + (1-t)\boldsymbol{v}$
$\quad \boldsymbol{c} \leftarrow \text{control}(\hat{\boldsymbol{\theta}}_1, \boldsymbol{x}_o)$
$\quad \tilde{\boldsymbol{v}} \leftarrow v_\phi^C(t, \boldsymbol{v}, \boldsymbol{c}) + \boldsymbol{v}$
$\quad u_t(\boldsymbol{\theta}|\boldsymbol{\theta}_1, \boldsymbol{x}_o) \leftarrow \frac{\boldsymbol{\theta}_1 - (1-\sigma_{\min})\boldsymbol{\theta}}{1 - (1-\sigma_{\min})t}$
$\quad \mathcal{L}_{\text{CFM}} \leftarrow ||\tilde{\boldsymbol{v}} - u_t(\boldsymbol{\theta}|\boldsymbol{\theta}_1, \boldsymbol{x}_o)||_2^2$
$\quad \theta \leftarrow \text{Update}(\phi, \nabla_\phi \mathcal{L}_{\text{CFM}}(\phi))$
**return:** $v_\phi, v_\phi^C$

---

**Algorithm 2** FM with Self-conditioning

**Input:** Training distribution $q_1$, flow network $v_\phi$, $\sigma_{\min}$
**while** Training **do**
$\quad (\boldsymbol{\theta}_1, \boldsymbol{x}_o) \sim q_1; \; z \leftarrow \mathcal{N}(0, I); \; s \leftarrow \mathcal{U}(0, 1)$
$\quad \boldsymbol{\theta} \leftarrow t\boldsymbol{\theta}_1 + (1-t)\boldsymbol{z}; \; \hat{\boldsymbol{\theta}}_1 \leftarrow 0$
$\quad \boldsymbol{v} \leftarrow \text{stopgrad}(v_\theta(t, [\boldsymbol{\theta}, \hat{\boldsymbol{\theta}}_1], \boldsymbol{x}_o))$
$\quad$ **if** $s > 0.5$ **then**
$\quad\quad \hat{\boldsymbol{\theta}}_1 \leftarrow \boldsymbol{\theta} + (1-t)\boldsymbol{v}$
$\quad \tilde{\boldsymbol{v}} \leftarrow v_\phi(t, [\boldsymbol{\theta}, \hat{\boldsymbol{\theta}}_1], \boldsymbol{x}_o)$
$\quad u_t(\boldsymbol{\theta}|\boldsymbol{\theta}_1, \boldsymbol{x}_o) \leftarrow \frac{\boldsymbol{\theta}_1 - (1-\sigma_{\min})\boldsymbol{\theta}}{1 - (1-\sigma_{\min})t}$
$\quad \mathcal{L}_{\text{CFM}} \leftarrow ||\tilde{\boldsymbol{v}} - u_t(\boldsymbol{\theta}|\boldsymbol{\theta}_1, \boldsymbol{x}_o)||_2^2$
$\quad \theta \leftarrow \text{Update}(\phi, \nabla_\phi \mathcal{L}_{\text{CFM}}(\phi))$
**return:** $v_\phi$

---

# B  SIMULATION-BASED INFERENCE

**Baselines comparison in section 5.1**  For a fairer comparison, we set up all baseline methods with a similar number of network weights and available compute time.

We train all baselines and flow matching with a batch size of 512 on the largest $10^5$ simulation budges for all tasks. For optimization, we apply Adam (Kingma & Ba, 2015) with default settings and constant learning rate of $10^{-4}$ and weight decay $2 \times 10^{-5}$.

All network architectures are chosen to have a similar number of ca. $3 \cdot 10^5$ parameters. For flow matching and continuous normalizing flows (CNFs), we use the same architecture based on a dense feed-forward neural net with skip connections using 8 residual blocks with each 128 neurons and *elu* activation. As input, we concatenate time $t$ and $\boldsymbol{\theta}_t$. For Neural Spline Flow (Durkan et al., 2019) and FFJORD (Grathwohl et al., 2019), we adopt the released implementation by the authors.

Depending on the time per epoch for each method, we modify the number of epochs and steps per epoch to allow all methods to train for a similar amount of time, ensuring a sufficient window for convergence. For NSF, we train for 1 000 epochs, for flow matching for 2 000 epochs, and for FFJORD and CNF 100 epochs.

**Flow matching with optimized hyperparameters**  For the experiments in section 5.2 and section 5.3, we adopt the hyperparameters and network architecture from Wildberger et al. (2023), which is based on a hyperparamter grid search. The hyperparameters for each task are listed in table 3. Otherwise, we follow the implementation as provided by the authors.

Table 3: Hyperparameters for SBI from Wildberger et al. (2023).

| Task | Time $\alpha$ | Batch size | Learning rate | Residual blocks |
|------|---------------|------------|---------------|-----------------|
| LV   | 1             | 32         | $10^{-3}$     | [32, 64, 128, 256, 5×512, 256, 128, 64, 32] |
| SLCP | -0.5          | 256        | $5 \cdot 10^{-4}$ | [32, 64, 128, 256, 5×512, 256, 128, 64, 32] |
| SIR  | 4             | 256        | $5 \cdot 10^{-4}$ | [32, 64, 128, 256, 7×512, 256, 128, 64, 32] |
| TM   | 4             | 64         | $2 \cdot 10^{-4}$ | [32, 64, 128, 256, 512, 3×1024, 512, 128, 64, 32] |

**Analyzing the 1-step estimate**  We simulate the flow ODE from the sampling distribution at $t = 0$ until $t^*$ ($x$-axis). Then, we compute the posterior in a single step by linearly extrapolating the flow, see eq. 6, to obtain the estimate $\hat{\boldsymbol{\theta}}_1$. Results are shown in fig. 8.

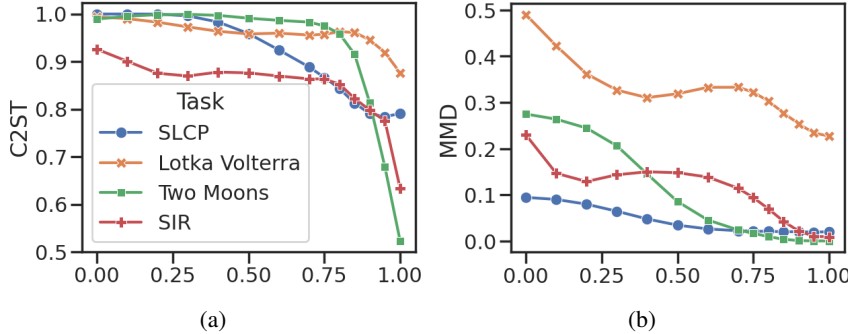

(a)  (b)

Figure 8: (a) and (b): C2ST score and MMD for predictive samples $\hat{\boldsymbol{\theta}}_1$. The $x$-axis shows from which we compute the predictive sample.

**Analyzing step size**   We analyze the influence of the step size of the ODE solver on the quality of the posterior distribution as shown in fig. 9.

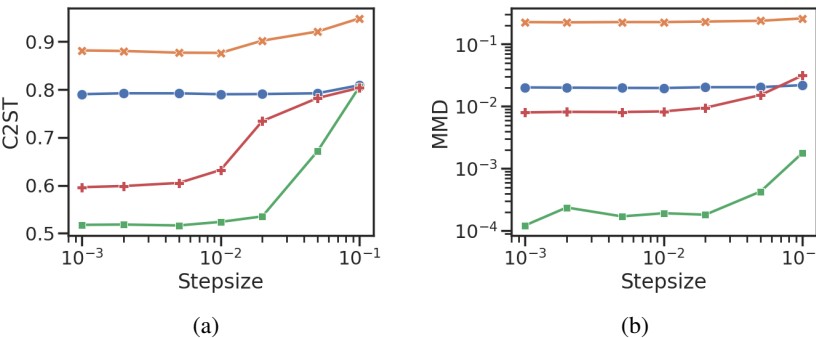

(a)                                                    (b)

Figure 9: (a) and (b): C2ST score and MMD vs. step size during inference.

**Additional results for maximum mean discrepancy**   For the evaluation in section 5.2, we show additional results for the maximum mean discrepancy (Gretton et al., 2012, MMD) in fig. 10.

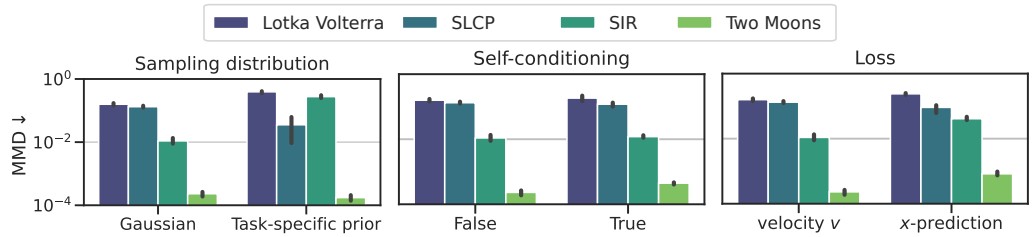

Figure 10: Evaluation of SBI tasks using different variants of flow matching training. Lower MMD scores are better.

### B.1   RECTIFIED FLOWS

The 1-step estimate $\hat{\theta}_1$ becomes more accurate and closer to the end point of the trajectory $\theta_1$ as paths become straighter. Rectified flows (Liu et al., 2023) have been proposed to learn a coupling between two distributions by solving a nonlinear least squares optimization problem. Flows can be recursively rectified, leading to increasingly straighter paths. We have trained the $k$-th rectified flow up to $k = 3$ following Algorithm 1 from Liu et al. (2023) for the Lotka Volterra SBI task. Networks, optimizers and learning rates are the same as for the flow matching experiments. Results for the rectified flows are show in table 4. The C2ST score gets worse for the 2- and 3-Rectified flow. We also finetune with gradient-based control signals. A difference compared to the finetuning experiments in section 5.3 is that we train the network starting at $t \geq 0$, whereas we have used $t \geq 0.8$ before. As flows become straighter, the 1-step estimates should become more reliable. This is why we consider finetuning with the control signal on the entire trajectory in this experiment, instead of only focusing on the last part ($t \geq 0.8$). When adding the finetuning, the C2ST score becomes better for the 1-Rectified flow compared to the 2-Rectified flow, indicating that the 2-Rectified flow produces more reliable 1-step estimates. However, the results for the rectified flows are not as good as the flow matching setup with $t \geq 0.8$ which we have used in section 5.3.

|  | ODE solution | + gradient-based controls |
|---|---|---|
| 1-Rectified Flow | 0.94 | 0.91 |
| 2-Rectified Flow | 0.98 | 0.89 |
| 3-Rectified Flow | 0.98 | 0.89 |

Table 4: C2ST score of the Lotka Volterra task for the $k$-th rectified flow.

## C  STRONG GRAVITATIONAL LENSING

We consider the following models:

- For modeling the lens we use an SIE model with 6 parameters: the Einstein radius $\theta_E$, the ellipticities $e_1$ and $e_2$ and $x_{\text{center}}$ and $y_{\text{center}}$. There is shear, for which we only consider $\gamma_1$ and $\gamma_2$ as free parameters.
- The source is modeled by a Sersic profile with free parameters being the amplitude, the half-light radius, the Sersic index $n$, the ellipticities $e_1$ and $e_2$ as well as the positions $x_{\text{center}}$ and $y_{\text{center}}$.
- The lens light is modeled in the same way as the source, although when generating the mock data, we fix the position as well as ellipticities to be the same as the lens mass model. For training and inference, we infer positions for both lens mass and lens light model, so the model could produce different values for them. The MCMC methods use the same parameter for both lens light and lens mass position.

We list all priors in table 5, table 6 and table 7. We do not have priors on the ellipticities $e_1$ and $e_2$ directly, but we obtain them from priors on the position angle and axis ratio. Also, we obtain the shear parameters from $\gamma_1$ and $\gamma_2$ from $\phi_{\text{ext}}$ and $\gamma_{\text{ext}}$ by converting them polar to cartesian coordinates. For SBI, we also include the two parameters $\text{ra}_0$ and $\text{dec}_0$ for the shear, which are always set to 0 when generating the training data sets, but in general our network could infer other values. Overall, there are 23 parameters for $v_\theta$, which fully describe the simulation setup. However, in our dataset there are only 17 free parameters. The MCMC methods only infer the reduced set of parameters, making use of the dependencies between them.

**Measurement instruments**  Observations have 160 times 160 pixels. The pixel size is 0.04 arc seconds. We use a Gaussian points spread function (PSF) with full width at half maximum (FWHM) of 0.3. The there is Gaussian background noise with a root mean-squared values of 0.01 and an exposure time of 1000s.

**Setup of MCMC-based methods**  We setup both baselines methods as follows:

1. Hamiltonian Monte Carlo: we use the No-U-Turn samples with a maximum tree depth of 10 and 5 000 warmup steps.
2. Affine-Invariant Ensemble Sampling: we use DEMove and StretchMove both with probability 0.5. There are 400 chains and we warm up for 20 000 steps before starting sampling.

Both methods are implemented in numpyro and optimized with JAX, so their runtimes are comparable with each other.

**Network architectures and training**

- Our flow network $v_\phi$ comprises a lightweight feature extraction network, represented by a CNN, which is consists of 6 downsampling blocks with 1 layer each a 32 channels and kernel size 3. As postprocessing of the output, we apply GroupNorm, silu and an additional 2DConv layer with kernel size 3 and a single channel. We reshape the output and feed it through a final dense layer. The output of the feature extraction has the same dimensionality as the parameters $\boldsymbol{\theta}$.
- An additional dense feed-forward neural network receives the concatenated the time $t$, $\boldsymbol{\theta}_t$ and extracted features as input. The feed-forward neural neural networks consists of 8 residual blocks with hidden dimension 128 and elu activation.
- The control network $v_\phi^C$ is represented by a small feed-forward neural network, consisting of 3 residual blocks with 64 hidden layers and 3 residual blocks with 32 hidden layers. We condition each block on the time via gated linear units and use a 16 dimensional time embedding.

For training, we use a batch size of 256 for the flow network $v_\phi$. When training $v_\phi^C$, we decrease the batch size to 16. We use the Adam optimizer with a learning rate of $10^{-4}$ and weight decay of $10^{-5}$.

Table 5: Priors for lens mass model parameters

| Parameter | Prior |
|---|---|
| $x_{\text{center}}$ | $\mathcal{U}(-0.2, 0.2)$ |
| $y_{\text{center}}$ | $\mathcal{U}(-0.2, 0.2)$ |
| position angle $\phi$ | $\mathcal{U}(0, 180)$ |
| axis ratio $q$ | $\mathcal{U}(0.25, 1)$ |
| external shear orientation $\phi_{\text{ext}}$ | $\mathcal{U}(0, 180)$ |
| external shear strength $\gamma_{\text{ext}}$ | $\mathcal{U}(0, 0.1)$ |
| Einstein radius $\theta_E$ | $\mathcal{U}(0.5, 2.0)$ |

Table 6: Priors for the source light

| Parameter | Prior |
|---|---|
| amplitude | $\mathcal{U}(5.0, 10.0)$ |
| half-light radius | $\mathcal{U}(0.5, 2.0)$ |
| Sersic index $n$ | $\mathcal{U}(1.5, 4.0)$ |
| position angle $\phi$ | $\mathcal{U}(0, 180)$ |
| axis ratio $q$ | $\mathcal{U}(0.25, 1)$ |
| $x_{\text{center}}$ | $\mathcal{U}(-0.2, 0.2)$ |
| $y_{\text{center}}$ | $\mathcal{U}(-0.2, 0.2)$ |

Table 7: Priors for the lens light

| Parameter | Prior |
|---|---|
| amplitude | $\mathcal{U}(5.0, 10.0)$ |
| half-light radius | $\mathcal{U}(0.5, 2.0)$ |
| Sersic index $n$ | $\mathcal{U}(1.5, 4.0)$ |

Training $v_\phi$ was done on a single NVIDIA Ampere A100 GPU for ca. 45 hours. We trained $v_\phi^C$ for an additional 24 hours. A lot of the training time was spent on running evaluation metrics, so it can be substantially improved.

C.1 DIFFUSION POSTERIOR SAMPLING (DPS)

We setup diffusion posterior sampling Chung et al. (2023a) as an additional baseline. The training dataset is the same as in 6, however since the diffusion model is unconditional, we drop any conditioning information.

**Network architecture and training** The neural network architecture is a multilayer perceptron MLP with 8 residual blocks and 128 neurons each. The activation function is *elu*. As optimizer, we use Adam with weight decay ($10^{-5}$). We train for 2000 epochs and for each epoch we sample 1000 batches from the dataset using a batch size of 4. We train the network as a denoising diffusion probabilistic model (DDPM) following Ho et al. (2020).

**Unconditional generation** Below, in figure 11, we visualize three samples generated by unconditionally sampling from the model. The observations are created using the lensing simulation code with the generated samples as input.

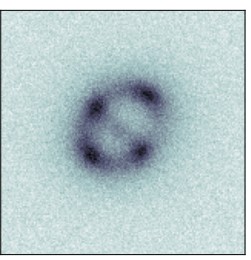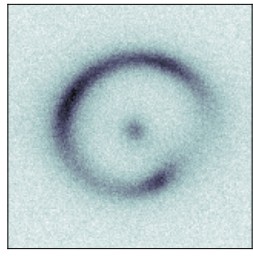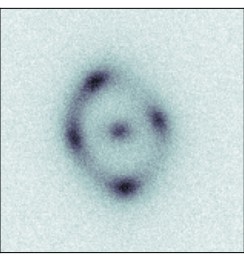

Figure 11: Visualization of unconditionally generated lensing systems.

**Inference** We directly follow Chung et al. (2023a) Algorithm 1 for inference, where the measurement operator $\mathcal{A}$ is replaced by the lensing simulation code. The step size in the algorithm is defined via a hyperparameter $\zeta$, which needs to be finetunes depending on the problem. We empirically test different values for $\zeta$ to find an optimal choice. Our results are shown in table 8. In this evaluation, we only model a smaller number of systems ($n = 25$).

| $\zeta$ | 0.0 | 0.0005 | 0.001 | 0.005 | 0.01 | 0.05 |
|---|---|---|---|---|---|---|
| Avg. $\chi^2$ | 28.15 | 16.20 | **9.98** | 10.07 | 12.98 | 12.64 |
| Min. $\chi^2$ | 15.14 | 3.84 | 3.07 | 1.58 | **1.40** | 1.53 |

Table 8: Evaluation of DPS and choosing $\zeta$.

C.2 LOSS-GUIDED DIFFUSION

We consider loss-guided diffusion (LGD) with a Monte Carlo-based estimate of the guidance term (Song et al., 2023, LGD-MC). LGD-MC can be seen as an extension of DPS, which uses $m$ points for estimating the guidance term, whereas DPS only uses a single point. We have evaluated LGD-MC using a different number of points $m$ using 100 steps for each sample. Because multiple points are used for the calculation of the guidance term at each step, the number of simulator calls grows by a factor of $m$. Results are shown in table 9 below. Similar to the DPS evaluation, we only consider a smaller subset of systems ($n = 25$). Interestingly, even though LGD can be seen as an extension to DPS, it performs worse. Ther performance of DPS critically depends on the hyperparameter $\zeta$ that corresponds to the step size and needs to be finetunes. LGD does not have this hyperparameter.

C.3 TWISTED DIFFUSION SAMPLER

Twisted diffusion sampler (TDS) is a sequential Monte Carlo algorithm for asymptotically exact conditional sampling from diffusion models that has been proposed by Wu et al. (2023). We evaluate

| $m$ | 5 |
| --- | --- |
| Avg. $\chi^2$ | 21.62 |
| Min. $\chi^2$ | 2.52 |
| Modeling time | $\sim 1600$s |

Table 9: Evaluation of LGD-MC for number of points $m$.

TDS using a different number of particles $K$ with the unconditional diffusion model from section C.1. We follow Algorithm 1 from Wu et al. (2023) using 100 steps ($T = 100$). In the paper, the algorithm is described for a variance exploding (VE) noise schedule. We adjust the algorithm for the variance preserving (VP) as described in Wu et al. (2023) Appendix A. We give results below in table 10. Similar to the DPS evaluation, we only consider a smaller subset of systems ($n = 25$).

| $k$ | 100 |
| --- | --- |
| Avg. $\chi^2$ | 20.94 |
| Min. $\chi^2$ | 2.47 |
| Modeling time | $\sim 210$s |

Table 10: Evaluation of TDS for number of particles.

## C.4 SIMULATION-BASED CALIBRATION

We use simulation-based calibration (Talts et al., 2018, SBC) as an additional evaluation method to assess the correctness of the posterior distributions. We adopt the SBC implementation from the Python package *sbi*. Below, in fig. 12, we show histograms for 8 parameters based on $n = 1000$ lens systems with $L = 1000$ posterior samples each.

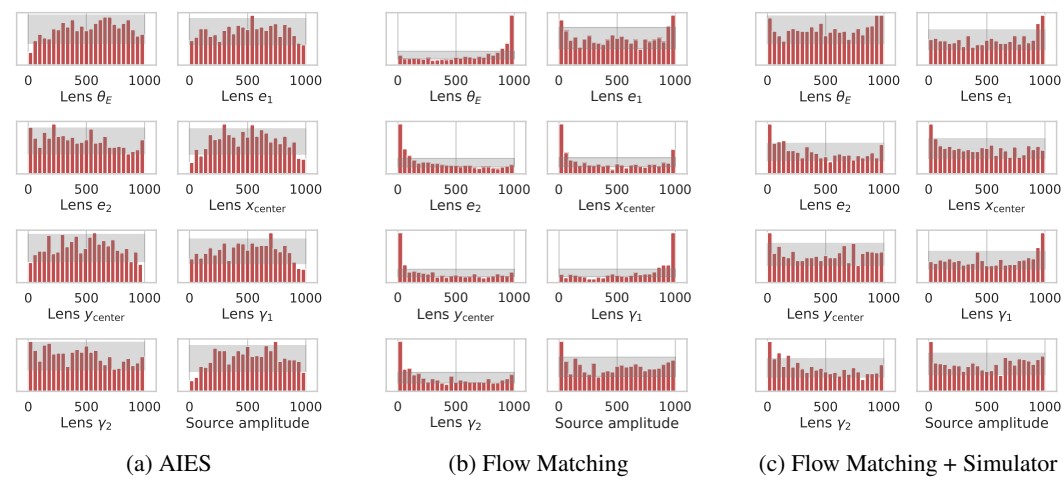

(a) AIES          (b) Flow Matching          (c) Flow Matching + Simulator

Figure 12: Simulation-based calibration histograms for different inference methods.

## C.5 TESTS OF ACCURACY WITH RANDOM POINTS

We have included an additional evaluation using sampling-based accuracy testing of posterior estimators (Lemos et al., 2023, TARP), see figure 13. We included HMC initialized with the ground truth values as a reference, which shows perfect coverage. If not initialized with the ground truth parameters, HMC and AIES produce biased samples. Flow matching more closely covers the posterior and shows visible improvements when including simulator feedback.

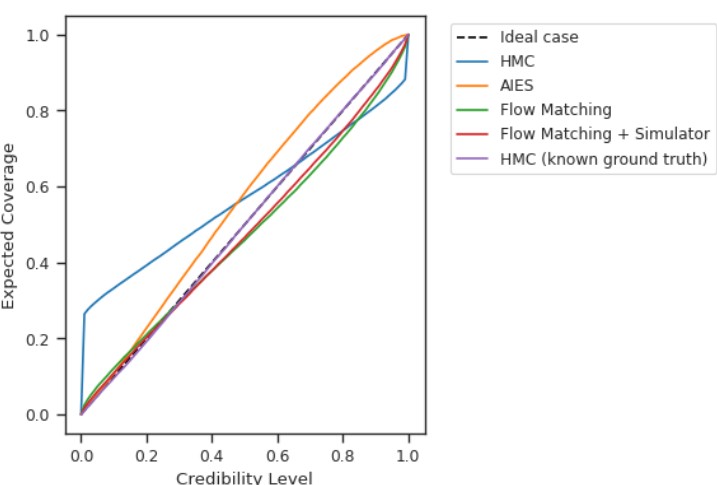

Figure 13: Evaluation of posterior coverage using TARP based on $n = 1000$ lens systems with $L = 1000$ posterior samples each.

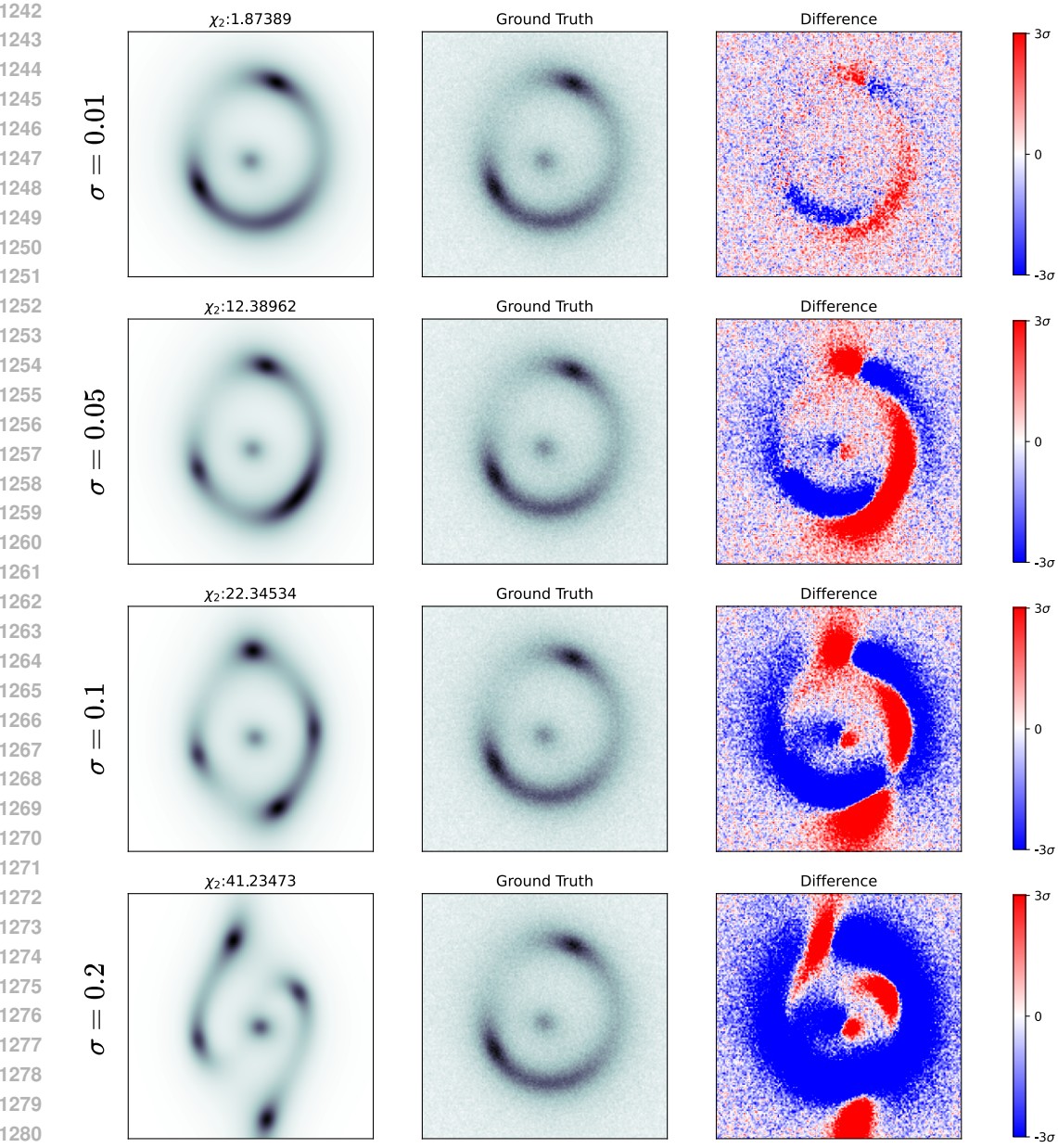

Figure 14: We show how a small noise $\sigma$ affects the simulated observation. We add a normal Gaussian with mean 0 and standard deviation $\sigma$ to a lens system's ground truth parameters $x$. Then, we plot the simulated observation based on the noised parameters and show the residuals.

## D    POSTERIORS AND RECONSTRUCTIONS FOR LENS MODELING

We show how small perturbations in the lens system's parameters affect the simulated observation in figure 14. We show extended plots of the posteriors in fig. 15 for lens system 1 and fig. 16 for lens system 6. Additionally, we show reconstructions based on flow matching with and without simulator feedback of lens systems 1 to 6 in fig. 17

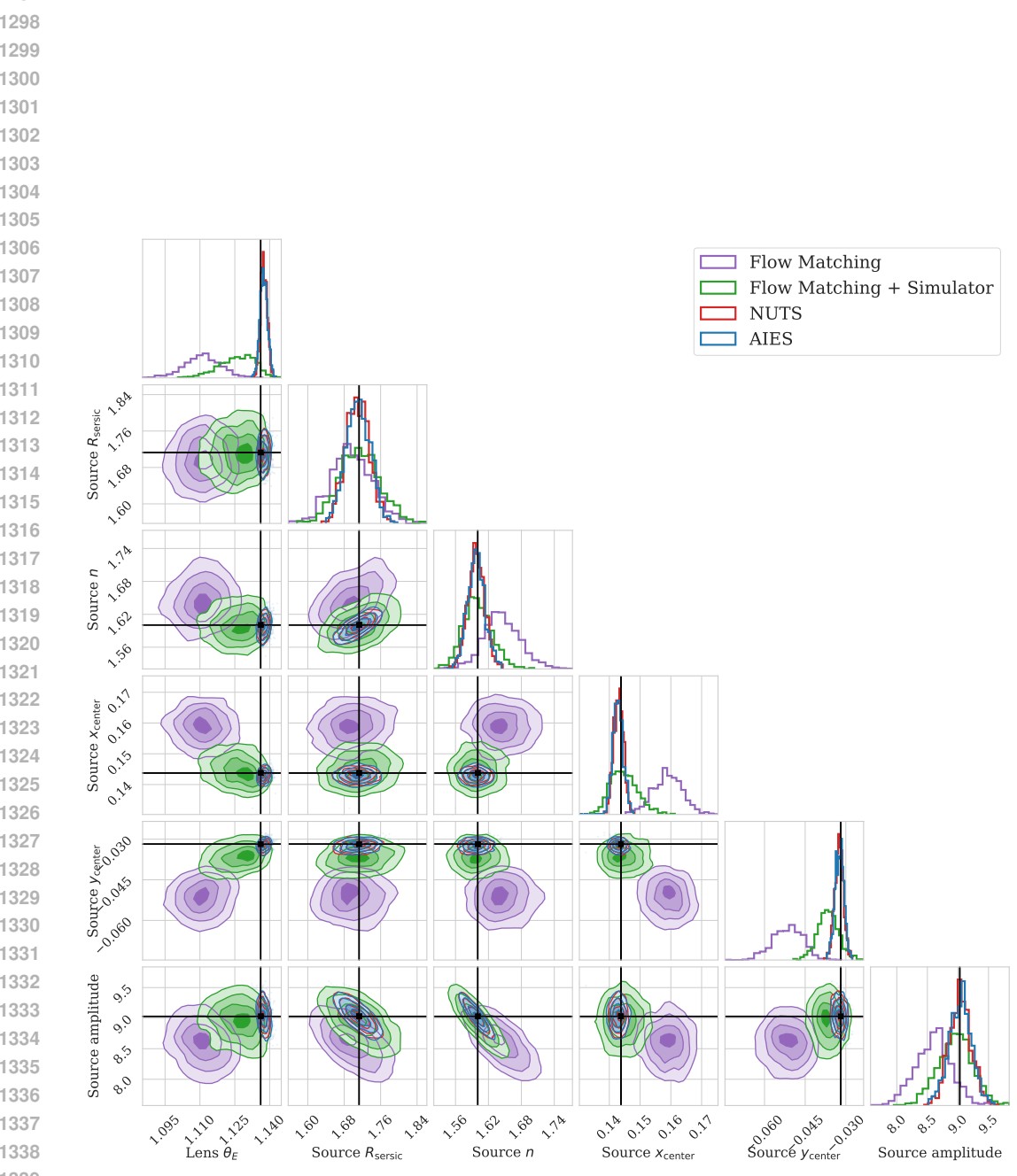

Figure 15: Posterior plot for system 1.

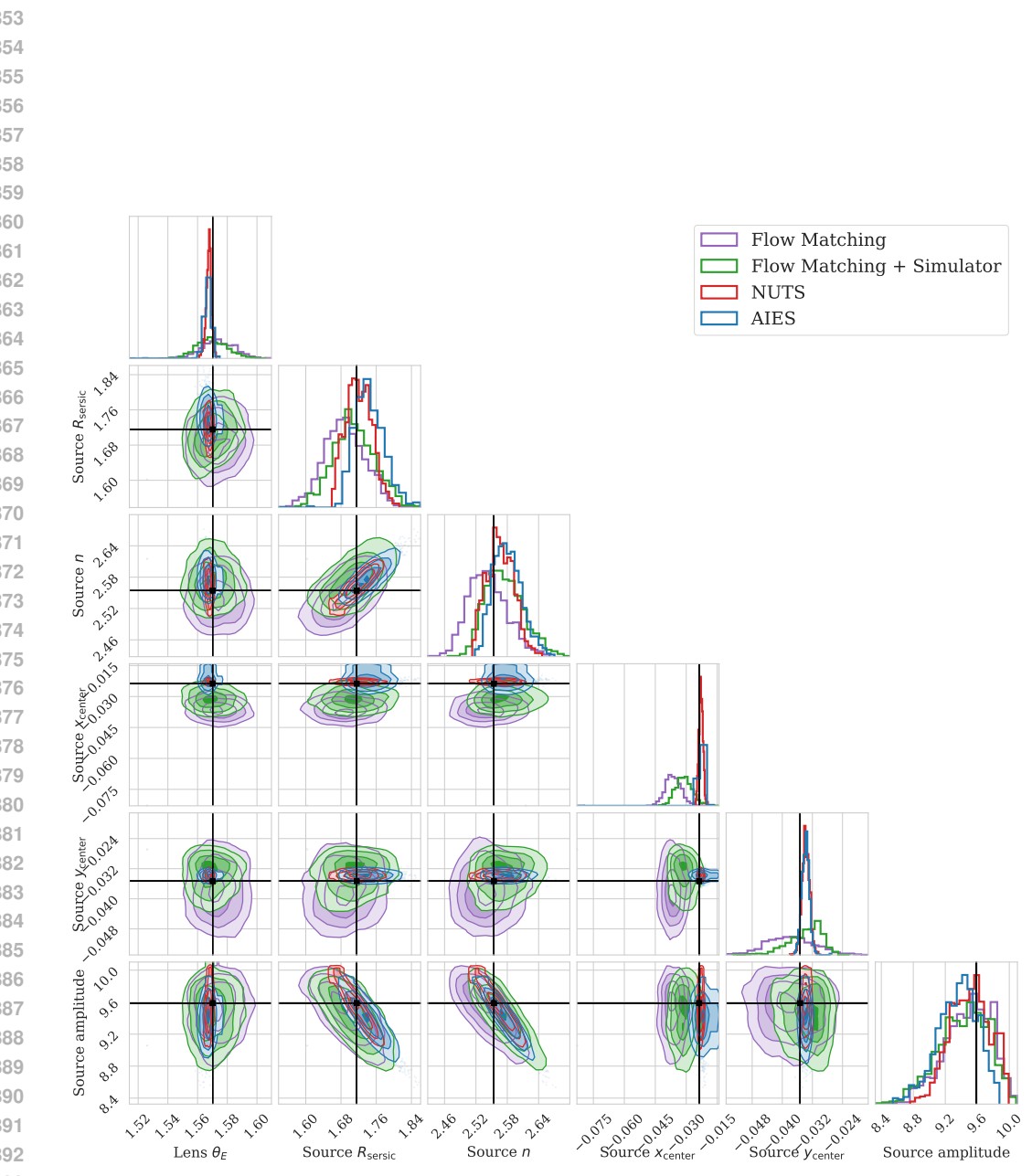

Figure 16: Posterior plot for system 6.

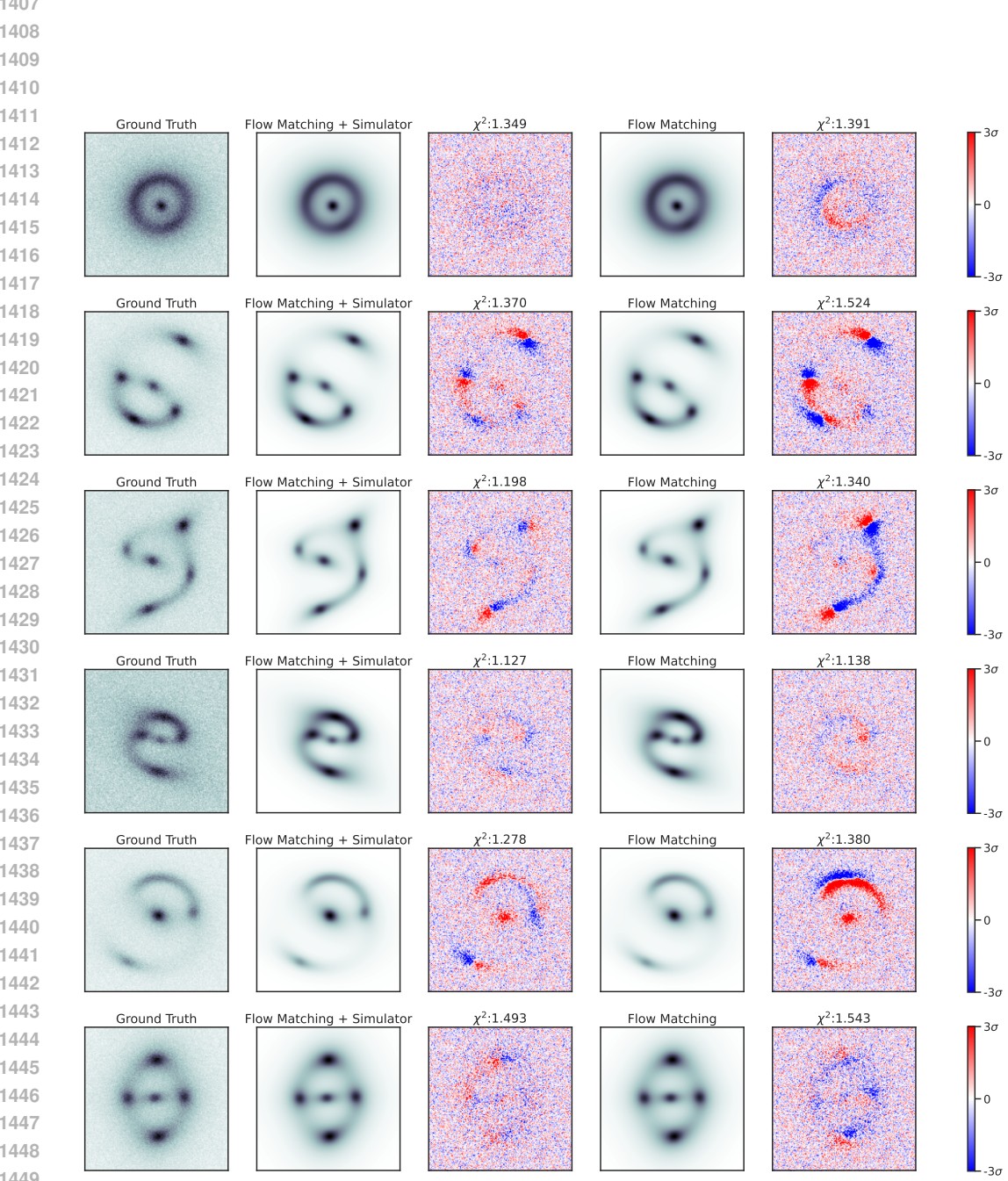

Figure 17: Modeling of different lens systems: system 1 (top) to system 6 (bottom).

