# OpenReview forum: "Flow Matching for Posterior Inference with Simulator Feedback"
_ICLR.cc/2025/Conference — Submitted to ICLR 2025_

### Official Review · Reviewer_ZUi2 · 2024-11-03

**Soundness:** 3
**Presentation:** 3
**Contribution:** 3
**Rating:** 6
**Confidence:** 4

**Summary:**

The paper presents a method for including simulator guidance in a flow matching-based simulation-based inference setup. The basic idea as far as I understand is to model the posterior using generative flow matching, and during training (possibly only at a fine-tuning stage), draw samples from the simulator by interpolating parameter samples from the velocity trajectory, and include an additional signal ensuring consistency between those samples and the training datapoint. The overall motivation is to increase simulation efficiency when a stochastic simulator is used for simulation inference (as is usually the case).

**Strengths:**

- Timely paper on an important problem in the field -- using information from the simulator to speed up simulation-based inference, in order to make it more sample-efficiency and well-calibrated.
- Discussion of several control mechanisms, which could be applicable depending on the specific simulator and domain.
- Good discussion of underlying theory, both the flow matching aspect as well as simulation-based inference. Sound comparison to surrounding context literature, e.g. relevance to classifier-free guidance.
- Application to an important and challenging problem in cosmology -- strong lens modeling, and comparison with a baseline HMC approach.

**Weaknesses:**

- In the strong lensing section, while a comparison with HMC is done, a comparison with "traditional" neural simulation-based inference approaches like NPE is lacking. This comparison would significantly strengthen the outcomes of this experiment.
- The pretraining/finetuning tradeoff is not comprehensively explored -- as far as I understand, a big advantage of the method is that one could just finetune using a smaller number of simulation calls. While this is mentioned briefly and tested for specific cases, a more comprehensive study of how fraction of finetuning for a fixed simulator budget affects the outcome would make the results quite a bit stronger.
- The high-level presentation could be made slightly clearer -- e.g., in Fig. 1, marking that the it is the posterior $p(\theta\mid x)$ that is being modeled/targeted, to immediately situate the reader to the problem setting.

**Questions:**

- Is there something particular about the flow matching setup (e.g. linear trajectories) that make simulator control applicable here, in contrast to a more traditional method like NPE? Does the method rely on assuming optimal transport coupling paths (eqs. 5-6), or is it more generally applicable?
- If I understand correctly, the goal of simulator control is to produce flow matching trajectories that better model the joint $(\theta, x_0)$ space by giving an additional signal that the interpolated parameter-generated samples should produce simulations consistent with the original $x_0$. This reduces the effect of simulator stochasticity. Could this partly be recreated by generating multiple samples from the same parameter point partway through training? Similarly, could producing a batch of samples for comparison to the $x_0$ further improve the flow matching control signal?
- In the strong lensing section, why was the comparison primarily made to HMC, rather than e.g. NPE?
- How critical was the $t > 0.8$ empirical threshold choice for controls, and do the authors expect this to be problem-dependent or fairly universal?

---

> ### Author Response · Authors · 2024-12-02
> **Response ZUi2**
>
> Thank you for the detailed review, thoughtful suggestions and positive feedback.
>
> - We agree that including more comparisons with traditional NPE methods would strengthen the empirical results. We have included two additional methods for conditional sampling from diffusion models (LGD-MC and TDS, see global response), but will add more in a future version of the paper. Additionally, we will explore the pretraining/finetuning tradeoff more comprehensively for the gravitational lensing experiment and update figure 1 based on your suggestion.
>
> Regarding your questions:
> - *Is simulator feedback limited to flow matching?* The optimal transport coupling paths that we have used are nice, because they produce straighter paths, thus improving the accuracy of the 1-step estimate. In principle, this can be replaced by other, similar objectives/couplings. As for whether simulator feedback can be used for more general NPE methods, we think it depends on the specific method and is hard to answer in general. Flow matching/diffusion can be used to give 1-step estimates, based on which we get feedback from the simulator to correct the current trajectory. For other NPE approaches, this mechanism for simulator feedback would need to be rethought.
> - *Improvements from multiple samples* Yes, that is a good idea. If we considered a batch of samples for comparison with the observation instead of a single sample, this can lead to a better simulator control (especially for stochastic simulators). We didn't consider it in this work, since it makes the training and inference more complicated, but it is a great idea for follow-up work.
> - *Comparison with other NPE methods* We will include more comparisons with NPE methods in an updated version of the paper.
> - *How critical was the $t>0.8$ empirical threshold?* We found it to be important, as it made it much easier for the control network to learn how to correct the flow based on the control signal. For $t < 0.8$ the 1-step estimates were not accurate enough so that feedback from the simulator was helpful. In this case, it was very difficult for the control network to extract useful corrections for the trajectory from the control signal. The threshold is directly related to how straight the flow paths are, which depends on both the problem and the flow matching setup.

---

> > ### Comment · Reviewer_ZUi2 · 2024-12-03
> >
> > Thanks to the authors for the response -- I will keep my 6 score, especially in absence of comparison with NPE benchmarks.

---

### Official Review · Reviewer_sNYF · 2024-11-04

**Soundness:** 2
**Presentation:** 2
**Contribution:** 2
**Rating:** 6
**Confidence:** 2

**Summary:**

This paper introduces a method to improve flow-based generative models for simulation-based inference by incorporating simulator feedback through control signals. The key idea is to refine a pretrained flow network with additional control signals based on simulator outputs, which can include gradients and problem-specific cost functions for differentiable simulators or learned signals from non-differentiable simulators. The authors demonstrate their method on several benchmark problems and show substantial improvements in accuracy (53%) while maintaining fast inference times (up to 67x faster than MCMC) on a challenging strong gravitational lensing application.

**Strengths:**

* The paper tackles an interesting problem of incorporating simulator feedback into generative models trained with flow matching. With the recent interest in the flow matching for learning generative models, the problem of incorporating downstream rewards, e.g. simulator feedback, is a critical one. This paper presents one of the first efforts in that direction (based on my knowledge).
* The application to the lensing problem is quite interesting. Generative models are well suited to such scientific inverse problems, and this method could be a useful addition to the toolbox for such problems.
* The method supports feedback from black-box simulators as well as differentiable simulators where gradient feedback is available.
* The method also provides considerable speed-ups in the simulation over other approaches.
* The paper is also quite clearly written and easy to follow.

**Weaknesses:**

* The empirical evaluation seems quite a bit limited. Specifically, the paper only considers a single applicaiton on strong gravitational lensing. The other synthetic tasks are quite small and it is unclear how general the method is. Moreover, the results on the gravitational lensing experiment, the results are not convicing. The residuals indicate the samples are not capturing the posterior. Finally, the authors consider a relatively simple variant of the problem (23D). In this setting FM + simulator feedback just acts as a faster approximation to MCMC. Where FM + Simulator feedback might have an advantage is high dimensional unstructured data (e.g. images in the case of lensing)
* Another shortcoming of the empirical evaluation is the relatively limited baselines. There are several approaches for unbiased inference with diffusion priors (like DPS) [1-4], so it would be good to add comparisons to some of these baselines.
* There is some missing discussion about related work about the guidance of flow matching models [5-6].
* The code to reproduce experiments is not provided though there are sufficient details in the paper.

[1] Wu, Z., Sun, Y., Chen, Y., Zhang, B., Yue, Y., & Bouman, K. L. (2024). Principled Probabilistic Imaging using Diffusion Models as Plug-and-Play Priors. arXiv preprint arXiv:2405.18782.

[2] Wu, L., Trippe, B., Naesseth, C., Blei, D., & Cunningham, J. P. (2023). Practical and asymptotically exact conditional sampling in diffusion models. Advances in Neural Information Processing Systems, 36.

[3] Dou, Z., & Song, Y. (2024). Diffusion posterior sampling for linear inverse problem solving: A filtering perspective. In The Twelfth International Conference on Learning Representations.

[4] Chung, H., Lee, S., & Ye, J. C. (2023). Decomposed Diffusion Sampler for Accelerating Large-Scale Inverse Problems. arXiv preprint arXiv:2303.05754.

[5] Nisonoff, H., Xiong, J., Allenspach, S., & Listgarten, J. (2024). Unlocking Guidance for Discrete State-Space Diffusion and Flow Models. arXiv preprint arXiv:2406.01572.

[6] Zheng, Q., Le, M., Shaul, N., Lipman, Y., Grover, A., & Chen, R. T. (2023). Guided flows for generative modeling and decision making. arXiv preprint arXiv:2311.13443.

**Questions:**

In addition to the weaknesses above:
* What are the challenges to scaling the approach to high-dimensional spaces?
* How sensitive is the method to the quality of the simulator? What happens when the simulator contains significant approximations or errors?
* Have you explored using multiple different types of control signals simultaneously? Could this provide complementary benefits?
* Could the method be extended to handle multiple observations simultaneously in a more efficient way?

---

> ### Author Response · Authors · 2024-12-02
> **Response sNYF**
>
> Thank you for the detailed review and very thoughtful suggestions.
>
> - **Limited empirical evaluation**: We agree that including more experiments can increase the quality of the empirical evaluation. Given the limited number of pages, it is however difficult to introduce and analyse multiple challenging real-world experiments and it was outside the scope of this rebuttal.
> - As mentioned in the general response, the gravitational lensing problem is not easy and increasing the dimensionality of the problem (for example by using a pixelated, high-dimensional representation of the source galaxy) also increases the degrees of freedom, which can actually make it easier for the predicted samples to faithfully reconstruct the observation. Therefore we think the dimensionality of the problem is at a sweet spot, where it is challenging, but can also be compared directly to MCMC approaches.
> - We have added two additional baselines (LGD-MC and TDS, see general response) which can be used for general non-linear inverse problems with a diffusion model prior.
> - We extended the related work to include the papers [5-6].
>
> Regarding your questions:
> - *What are the challenged to scaling the appraoch to high-dimensional spaces?* We think that flow matching with our proposed simulator feedback can scale to high-dimensional spaces in the same way that diffusion models/flow matching do.
> - *How sensitive is the method to the quality of the simulator?* If the simulator contains significant approximation errors that can be a problem. We want to improve the predicted posterior from flow matching using feedback from the simulator. However, if the simulator is not accurate, then feedback from the simulator might not provide useful information to further improve the samples. In this case, the control network will learn to ignore simulator feedback and output the pretrained flow without any corrections.
> - *Have you explored using multiple different types of control signals simultaneously? Could this provide complementary benefits?* We have not explored using multiple control signals simultaneously, but it is an interesting interesting idea and a good experiment for future work. We believe that there can be benefits from using multiple control signals in the same way that multiple, possibly physics-informed losses can help in many problems.
> - *Could the method be extended to handle multiple observations simultaneously in a more efficient way?* We have not done any experiments with multiple observations. It should be fairly strightforward to modify the control signals to account for multiple observations. The conditioning of the flow network for multiple observations can be more difficult. We think that an approach similar to [7] might work for conditioning flows as well, but we have not done any tests.
>
> [7] Geffner, T., Papamakarios, G., & Mnih, A. (2023). Compositional score modeling for simulation-based inference. In International Conference on Machine Learning (pp. 11098-11116). PMLR.

---

> > ### Comment · Reviewer_sNYF · 2024-12-02
> >
> > Thank you for the response and additional results in the rebuttal. I still believe that for a largely empirical paper, the experiments are limited (in terms of the scale as well as breadth) but the ideas presented are interesting for the community. I have raised my score to reflect this.

---

### Official Review · Reviewer_B78P · 2024-11-04

**Soundness:** 1
**Presentation:** 2
**Contribution:** 1
**Rating:** 3
**Confidence:** 5

**Summary:**

The paper considers the problem of solving inverse problems mostly in physical sciences. In particular, we are interested in posterior inference. This paper propose to use the flow matching perspective refined with additional control signals coming from a simulator. Moreover, the authors consider various scenarios depending on the differentiability of the simulator. Finally, they show the empirical results on a few simulator-based inference problems and the gravitational lensing inverse problem.

**Strengths:**

The paper has some strengths overall, which I will outline below.


**Strengths:**
1. Introduction of the flow matching perspective for these set of problems making a faster inference procedure.
2. Providing the solutions on using either differentiable or non-differentiable simulators.

**Weaknesses:**

Despite its strengths, the paper has a few major and minor weaknesses.

**Major weaknesses:**

1. I’m really concerned about the presentation of the results and the abilities of this method, because of the lack of enough comparisons and metrics.
2. Regarding the gravitational lensing problem (presented as the main real-world task being tackled), they are operating in a relatively small space, making the problem easy and solvable. However, they didn’t include any coverage test (e.g., TARP [1], or any other), so it’s hard to say if the posteriors are good. Moreover, as far as I understand, the evaluation test is only on their own simulations, the one that the model were being trained on. In particular, we don’t know if the model is robust to any OOD examples. Finally, the presented residuals (e.g., in Fig. 6) are looking bad.
3. Regarding the part: „(…) however, previous methods are usually restricted to point estimates, use simple variational distributions or Bayesian Neural Networks (Schuldt et al., 2021; Legin et al., 2021; 2023; Poh et al., 2022) that are not well suited to represent more complicated high-dimensional data distributions.” - Legin et al. 2021; 2023 use a likelihood-free inference (or simulation-based inference) framework to get posteriors from simple feed-forward nn (not Bayesian).
4. The authors proposed the intensive tests only on LV, which is relatively simple problem and for sure not enough for fair comparison. Moreover, the results in Tab. 1 don’t show any superiority of the proposed method - in particular, NSF is getting similar or better results.


**Minor weaknesses:**

1. The authors should include comparisons with other novel posterior sampling baseline methods than DPS, e.g., LGD−MC [2].
2. In line 319, should be „spline”


**References:**

[1] Lemos, P., Coogan, A., Hezaveh, Y., & Perreault-Levasseur, L. (2023, July). Sampling-based accuracy testing of posterior estimators for general inference. In International Conference on Machine Learning (pp. 19256-19273). PMLR.

[2] Song, J., Zhang, Q., Yin, H., Mardani, M., Liu, M. Y., Kautz, J., ... & Vahdat, A. (2023, July). Loss-guided diffusion models for plug-and-play controllable generation. In International Conference on Machine Learning (pp. 32483-32498). PMLR.

**Questions:**

I would like to see especially the experiments and responses to the issues mentioned as weaknesses.

---

> ### Author Response · Authors · 2024-11-29
> **Response B78P**
>
> Thank you for the detailed review and very thoughtful suggestions.
>
> - **Additional baselines**: We have added comparisons with Twisted Diffusion Sampler TDS [1] and Loss-Guided Diffusion LGD-MC [2] in table 2 as recommended by reviewers sNYF and B78P. Both methods do not produce posterior samples with accurate reconstructions of the observation and face similar difficulties as Diffusion Posterior Sampling (DPS).
> - **Coverage tests**: We have included the coverage test TARP [3] in figure 13, which was recommended by you. The evaluation shows that the flow matching (+simulator) posteriors have good coverage, albeit not perfect. They perform better than the MCMC-based baselines HMC and AIES for the given computational budget.
> - We have updated the related work section, adding a an extended discussion of Legin et al. (2023).
>
> We have addressed the concers regarding the flow matching results for the SBI tasks in table 1 and the posteriors/residuals of the gravitational lensing experiments in our global response.
>
> [1] Wu, L., Trippe, B., Naesseth, C., Blei, D., & Cunningham, J. P. (2023). Practical and asymptotically exact conditional sampling in diffusion models. Advances in Neural Information Processing Systems, 36.
>
> [2] Song, J., Zhang, Q., Yin, H., Mardani, M., Liu, M. Y., Kautz, J., ... & Vahdat, A. (2023, July). Loss-guided diffusion models for plug-and-play controllable generation. In International Conference on Machine Learning (pp. 32483-32498). PMLR.
>
> [3] Lemos, P., Coogan, A., Hezaveh, Y., & Perreault-Levasseur, L. (2023, July). Sampling-based accuracy testing of posterior estimators for general inference. In International Conference on Machine Learning (pp. 19256-19273). PMLR.

---

> > ### Comment · Reviewer_B78P · 2024-12-02
> >
> > Many thanks for the revised version of the paper, including additional coverage tests and another baselines. However, I still believe that this paper needs substantial improvement, so I will maintain my current score.

---

### Official Review · Reviewer_qrNo · 2024-11-04

**Soundness:** 1
**Presentation:** 2
**Contribution:** 2
**Rating:** 3
**Confidence:** 3

**Summary:**

This paper introduces flow matching with simulator feedback for simulation-based inference which extends offline flow matching for neural posterior estimation with an online phase that uses online simulations to improve the accuracy of the estimated posterior distribution. Indeed, the authors observe that learning perfectly the score function corresponding to the true posterior distribution is challenging and propose to access the simulator online and correct these imperfections at evaluation time. The method is supposedly much more efficient than alternative online method such as MCMC while having the potential to provide as accurate results. The paper introduces two types of control signals, a gradient-based and a learning-based control signal, which are made for differentiable (and deterministic) and non-differentiable (and potentially stochastic) simulators respectively. The method is empirically tested on 4 common SBI benchmarking tasks and a "strong gravitational lensing" problem. Results highlight that simulator feedback help improving the accuracy of the posterior distributions.

**Strengths:**

- *Novelty*: To the best of my knowledge, the idea of including simulator feedback in flow matching for posterior estimation is novel
- *Soundness*: The idea of using simulator feedback to correct for the imprecision of score matching may indeed be helpful in certain applications.
- *Presentation*: I found the tables and figure easy to read and informative, while being also visually appealing.

**Weaknesses:**

While I must acknowledge certain positive aspects of the paper, I have also several concerns that motivate my negative recommendation, which I am listing below.
1. While I find the figures and tables quite enlightening, I find the presentation being a weakness. There are multiple hand-wavy explanations and claims that I find a bit confusing. See below for concrete examples.
2. Empirical validation: I was surprised by the numbers from figure 3 and 4 which seems worse than existing alternatives. In particular, the paper only compares to offline methods while it is clear from [1] that SNLE or SNPE which are sequential alternatives to NPE/NLE and perform much better than the proposed method. It is also arguable whether these benchmarks are the most relevant ones as there are quite low dimensional in term of observation dimensionality and are not fully representative of applications where SBI can shine. For instance, gravitational waveforms and spiking neurons are interesting benchmark used in many SBI papers. I also find the results of Flow matching + posterior quite bad compared to MCMC methods in figure 14 and 15 where we can clearly see an issue with bias and also variance of the predicted posteriors.
Overall, the empirical validation seems insufficient to me and do not clearly demonstrate that the method proposed is of any real use.
3. Relevance: Sequential SBI methods, which call the simulator online, are often complicated to motivate as they require to access the simulator while also arguing doing inference on this simulator without SBI is hard because evaluating the simulator is computationally demanding. While I agree this is not the case for all applications, I would expect the paper to clearly highlight and benchmark the method on use cases where simulating more samples offline to get a good amortised posterior is not enough and calling the simulator online does not take too much time.

I am not confident these concerns can be addressed in the scope of the discussion (especially regarding results and presentation) but I am open to discussing with authors.

Hand-wavy explanations examples:
- l33-36: Seems an intricated way of explaining likelihood and posterior distributions which are very standard mathematical objects most reader should already know about.
- l37-43: The presentation of SBI is again a bit weird in my opinion. It does not clearly say when and why SBI may be necessary and what it solves. It may be interpreted as if SBI was only about Bayesian inference for the uninformed reader where frequentist methods exist as well.
- l48: what do you mean by "it became clear ... be specified a priori"?
- l49-50: this a vague and pretty strong statement that I would expect to be clarified and supported by reference.
- l52-54: I do not understand what you mean by saying there is "no feedback loop".
- l74-78: you mix the high level description of the method with specific implementation decisions which makes it hard to understand on what aspects the reader should focus on.
- Section 3 was presented in a way that I did not find easy to digest.
- l205: "a fundamental problem..." why is it a fundamental problem is unclear to me.
- l254-255: What shall we conclude from that sentence?
- l277-282: It seems like a hacky solution
 - l283: While I understand you are trying to emphasise that the algorithm still learn the "true" posterior distribution, this is stated in a hand-wavy way in my opinion.
- l345-351: This is quite confusing again. Why is there a dropout layer? It is quite complicated to grasp the details of all variants.
- 5.4: This is again a very hand-wavy claim without actually being theoretical or empirical support.


[1]:http://proceedings.mlr.press/v130/lueckmann21a/lueckmann21a.pdf

**Questions:**

I would be happy if authors could provide arguments against my concerns.

---

> ### Author Response · Authors · 2024-11-29
> **Response qrNo**
>
> Thank you for the detailed review and very thoughtful suggestions.
>
> We have updated the manuscript based on your feedback and suggestions.
>
> - **Sequential NPE/NLE methods** We focused on non-sequential NPE methods in this paper, however we agree with you that more baseline comparisons with sequential SBI methods is a good suggestion. If the 1-step estimates become better as paths become straighter (for example due to further improvements in the flow matching training, rectified flows, consistency models, better couplings, etc., then the 1-step estimate becomes exact. Therefore inference via the ODE with simulator feedback can be seen as a sequential method, as predictions get refined in each step with simulator feedback. This sequential approach avoids many complications of other sequential SBI methods such as mismatches with proposal posteriors, truncations or the need to retraining networks on new samples during inference.
> - **More experiments** We agree that experiments on gravitational waveforms and spiking neurons are interesting, but we think that they are outside of the scope of this rebuttal. We want to stress again that while the dimensionality of the gravitational lensing problem is low for the parameters we want to infer, the dimensionality of the observation is high (a 160 x 160 image). Calling the simulator online is relatively cheap, but it is very sensitive to several parameters, which makes the problem difficult and a nice "challenge" for SBI.
> - **Bad posterior compared to MCMC and more baselines**
> We have updated figures 15 and 16 which now also show the posteriors obtained without simulator feedback. As can be seen, including simulator feedback visibly and consistently reduces the bias of the posterior. We have also run an additional coverage statistic. The evaluation shows that the flow matching (+simulator) posteriors have good coverage, albeit not perfect. They perform better than the MCMC-based baselines HMC and AIES for the given computational budget. HMC and AIES do not always give the same posterior, as they may fail to converge sometimes when modeling a large number of systems, which explains why they compare worse than flow matching on average, but can perform better for specific systems such as the one shown in figure 15. Additionally, we have included more baselines for posterior sampling with diffusion models as mentioned in the general response, which demonstrate that simulator feedback gives a good solution for a difficult problem and outperforms other state-of-the-art methods for conditional sampling from diffusion models.

---

> > ### Comment · Reviewer_qrNo · 2024-12-03
> >
> > I appreciate the time you have spent in the rebuttal and believe the idea presented in your paper is interesting.
> >
> > Nevertheless, the paper requires major updates to demonstrate the value of the proposed approach. In particular, I encourage the authors to align the targeted applications, where their method would be valuable, with benchmarks that falls under that scope. I would encourage authors to demonstrate that their method is easily applied in such settings, saves compute, and performs on-par or better than existing alternatives.
> >
> > I have also checked other reviews and it seems that some of my concerns are shared with other reviewers.

---

### Official Review · Reviewer_tcVJ · 2024-11-06

**Soundness:** 3
**Presentation:** 2
**Contribution:** 2
**Rating:** 3
**Confidence:** 4

**Summary:**

The paper studies the problem of modelling the posterior $p(\theta\mid x)$ in a generative model $\theta\to x$, where $p(x\mid\theta)$ is available as a simulator, but possibly without access to exact likelihoods. When the sampling of $\theta$ given $x$ is modelled as a conditional (on $x$) neural ODE and this ODE is trained by flow matching objectives, it is proposed to place an inductive bias on the drift model: the output of simulator or its gradient, evaluated at an intermediate time point or its extrapolation to the target space, is encoded and given as an input to the drift model. Such a form of the drift model is hypothesised to improve the approximation of the target distribution by effectively guiding the drift to the modes of the posterior. Experiments are done on several low-dimensional simulation-based inference tasks, including the lens and source parameter estimation problem in strong gravitational lensing.

**Strengths:**

- The problem studied is highly relevant as foundation models (including diffusion and flow-based models) become available in various scientific domains. It is important to develop  inference methods for inverse problems that have low bias, good posterior coverage, and high efficiency of training and inferences -- this paper attempts to solve these problems.
- The proposed algorithm plausibly attacks these challenges (even if it is not very well demonstrated by the experiments, see below) and should give an asymptotically correct solution to the posterior sampling problem.
- Interesting analysis of algorithm variants in Section 5.2.

**Weaknesses:**

Throughout the text, there are many inaccurate or somewhat sloppy statements and references that confuse a specialist in flow matching models (and would likely impede understanding by non-specialists as well). A list follows.

- Abstract: I would suggest to revise it to explain the problem setting and approach at a higher level.
  - First sentences: "Flow-based generative modeling is a powerful tool for solving inverse problems in physical sciences" -- this is a bold claim. The use of flow-based models for inverse problems is not yet well-established; in fact, this is what this paper aims to do.
  - The next few sentences do not set up the problem well (it is not even explained that we are talking about continuous normalising flows).
  - The results at the end do not make sense without context: what does "improves the accuracy by 53%" mean?
- Introduction:
  - L046: "normalising flows transform a noise distribution to the posterior distribution" is true, but:
    - It is a statement with low specificity (VAEs and GANs also transform noise to data).
    - These models are first introduced in the setting of training from samples, which is not what we usually have in Bayesian inference (no ground truth samples from the posterior).
    - Two of the three citations about diffusion models are actually about ODEs / flow matching. The two are of course connected, but I think it is unfair to call flow-based models an instance of "success of diffusion models [...] specifying a corruption process". For instance, flow ODEs can be learned that are not the probability flow ODEs of diffusion processes, including Liu et al.'s rectified flow (the first iteration is indeed a diffusion ODE, the later 'straightened' ones are not) and Tong et al.'s minibatch OT-based flow matching.
      - One solution could be to explicitly state the connection between FM and diffusion in the cases where it exists (e.g., for Ornstein-Uhlenbeck noise, optimal drifts of ODE and SDE are both expressed in terms of the score, so learning one is tantamount to learning the other).
  - In the paragraph starting L052, somehow we have jumped from a general distribution-matching setting (which is not how flow-based models were introduced -- they are trained from samples!) to a conditional posterior modelling setting. Please state the setting/assumptions (e.g., that we have conditional posterior samples from a simulator).
- Related work:
  - "Inverse problems with diffusion models": This seems to be better named "solving inverse problems under a diffusion model prior". Although the works that do this with Monte Carlo (e.g., Cardoso et al, Dou et al,) are mentioned (you could also consider [Song et al.](https://proceedings.mlr.press/v202/song23k.html)), there is also stochastic optimisation (e.g., [Mardani et al.](https://arxiv.org/abs/2305.04391), [Graikos et al.](https://arxiv.org/abs/2206.09012)), or amortisation by RL methods (e.g., [Black et al.](https://arxiv.org/abs/2305.13301), [Fan et al.](https://arxiv.org/abs/2305.16381), [Venkatraman et al.](https://arxiv.org/abs/2405.20971)).
  - "Flow matching": It is strange to see "optimal transport paths (Lipman et al.)" contrasted with "independent couplings or rectified flows (Liu et al., Tong et al.)". In fact, it is Rectified Flow and OT-CFM (Liu et al., Tong et al.) who consider **non-independent couplings** through rectification steps or OT couplings (thus actually approximating the dynamic OT), respectively, while Lipman et al.'s flow matching is equivalent to one using independent couplings and is OT only on the level of the conditional probability paths used for training.
- FM theory:
  - Equation (1): $\theta$ in subscript should be $\phi$.
  - L156: Because smoothness conditions are stated, they should be precise: Do you assume Bochner integrability? Continuous differentiability (how many times?) in both $\theta$ and $t$? It should also be stated that $p_t(x)=p(t,x)$ and $p$ is a function $[0,1]\times\mathbb{R}^d\to\mathbb{R}$.
  - L182 is hard to understand: what is meant by "$q(z)=p_1(\theta)$? It should be said that the conditioning variable $z$ is identified with the endpoint $\theta_1$, etc.
- Controls for improved accuracy:
  - LL201-203 do not make sense to me. The paragraph begins with conditioning of ODEs -- how is an old trick in diffusion "for example" w.r.t. such conditioning?
  - NB. Equation (7) will be the *exact* $t=1$ endpoint of integration ($\theta_1=\hat{\theta}_1$) if we have a perfectly fit OT or any model with straight integral curves (such as the converged ODE after many iterations of rectified flow)!
  - LL225-227 and later at LL352-355: Once again, I am surprised by the repeated discussion of Liu et al. and Tong et al. yet the *omission of the actual algorithms they propose* (rectification and MBOT coupling), which both produce **straighter** paths than a vanilla FM (and hence inference in fewer steps).

Experiment results are not convincing:
- Results are not consistently showing improvement (cf. Table 1) and error bars are not reported, making it impossible to assess significance.
- In Section 5, can you comment on computational efficiency in terms of wall time?
- Lensing:
  - The evaluations do not seem to guarantee coverage ($\chi^2$ is obviously not sufficient for this, and the SBC tests use projection only on a single parameter). Have you considered coverage tests such as those used in Legin et al., Section 3? Currently it is not demonstrated that the proposed method achieves more accurate posterior sampling.

**Questions:**

Please see above.

Simpler than the encoder architecture, did you consider physics-inspired ways of providing the simulator information to the drift model? For example, simply expressing the drift as (learned vector field NN) + (learned scalar or diagonal NN) x (simulator gradient), as often done in work on diffusion models for sampling of Boltzmann distributions.

---

> ### Author Response · Authors · 2024-11-29
> **Response tcVJ**
>
> Thank you for the detailed review and very thoughtful suggestions.
>
> We have updated the manuscript based on your feedback and clarified the points that you had listed.
>
> - **Results are not consistently showing improvements (cf. table 1)**
> The experiments in table 1 are all low-dimensional with strong baselines producing good posteriors given the available dataset sizes/simulator budget. So flow matching being comparable to the best performing models is a good outcome, as flow matching/diffusion training have the advantage of scalability to larger networks and more high dimensional inputs. As mentioned in the general response, flow matching has been evaluated against other neural posterior estimation problems for these tasks already in [1]. Since we do follow-up experiments with modifications of flow matching in section 5, we have also included some baselines comparisons for the tasks that we had run ourselves.
> - **Computational efficiency in terms of wall time**
> Training and evaluating the test metrics on the flow matching network on the largest budget of $10^7$ simulations took ca. 52 minutes (200 epochs), however the generation of the dataset took an additional estimated x minutes. The network on $10^5$ simulations took on average ca. 13 minutes for training and evaluation. Finetuning with the gradient-based control signals took an additional 2 hours, 2 minutes (early stopping after 83 epochs) while with the learning-based control signal, the run took ca. 1 hour 32 minutes (early stopping at 150 epochs).
> - **Coverage tests using TARP [1]**
> We have included an evaluation of the coverage in appendix C.3, which shows that flow matching both with and without simulator feedback achieves good - although not perfect - results. They perform better than the MCMC-based baselines HMC and AIES for the given computational budget.
> - **Rectification of flows**
> We had cited the rectified flow paper by Liu et al. without discussing the rectification algorithm, as pointed out by you. It is a great idea to finetune with simulator feedback on the rectified flow, as paths should be straighter, thus producing better 1-step estimates.
> We have included experiments with the LV task in appendix B.1. We find that algorithm 1 from Liu et al. produces slightly worse results than the flow matching training following Lipman et al. When rectifying the flow in 2- and 3-Rectified Flow, the C2ST score gets worse. We think that the rectified flows perform worse, because of the conditioning on the observations and training on paired data in the reflow stage. When finetuning with simulator, the C2ST score for 2- and 3-Rectified Flows is better than for the finetuned 1-Rectified Flow, showing that with straighter paths, simulator feedback is better. However, because the velocity model is worse for 2- and 3-Rectified Flows, the final C2ST is not better than the one with the simulator feedback finetuning strategy used in section 5.3.
> - **Using simple control networks**
> You asked the question, if it is possible to use a simpler control network in the form a learned scalar or diagonal neural network times the simulator gradient. We had initially used simpler control networks, like the ones you mentioned. However we found in the gravitational lensing experiments that both the time and the value of the cost function provide significant improvements and very simple models did not work very well. We haven't considered a diagonal neural network, but it could be an interesting future experiment.
>
> [1] Wildberger, J., Dax, M., Buchholz, S., Green, S., Macke, J. H., & Schölkopf, B. (2024). Flow matching for scalable simulation-based inference. Advances in Neural Information Processing Systems, 36.

---

### Comment · Area_Chair_pxQg · 2024-11-26

Dear all,

The deadline for the authors-reviewers phase is approaching (December 2).

@For reviewers, please read, acknowledge and possibly further discuss the authors' responses to your comments. While decisions do not need to be made at this stage, please make sure to reevaluate your score in light of the authors' responses and of the discussion.

- You can increase your score if you feel that the authors have addressed your concerns and the paper is now stronger.
- You can decrease your score if you have new concerns that have not been addressed by the authors.
- You can keep your score if you feel that the authors have not addressed your concerns or that remaining concerns are critical.

Importantly, you are not expected to update your score. Nevertheless, to reach fair and informed decisions, you should make sure that your score reflects the quality of the paper as you see it now. Your review (either positive or negative) should be based on factual arguments rather than opinions. In particular, if the authors have successfully answered most of your initial concerns, your score should reflect this, as it otherwise means that your initial score was not entirely grounded by the arguments you provided in your review. Ponder whether the paper makes valuable scientific contributions from which the ICLR community could benefit, over subjective preferences or unreasonable expectations.

@For authors, please respond to remaining concerns and questions raised by the reviewers. Make sure to provide short and clear answers. If needed, you can also update the PDF of the paper to reflect changes in the text. Please note however that reviewers are not expected to re-review the paper, so your response should ideally be self-contained.

The AC.

---

### Author Response · Authors · 2024-11-29
**Global Response**

We thank all reviewers for their constructive feedback and we have updated the manuscript. The main changes comprise the following:

- **Updated text**: We have updated the text to clarify and improve some smaller issues mentioned by reviewers tcVJ, qrNO and B78P.
- **Additional baselines**: We have added comparisons with Twisted Diffusion Sampler TDS [1] and Loss-Guided Diffusion LGD-MC [2] in table 2 as recommended by reviewers sNYF and B78P. Both methods do not produce posterior samples with accurate reconstructions of the observation and face similar difficulties as Diffusion Posterior Sampling (DPS).
- **Coverage tests**: We have included the coverage test TARP [3] in figure 13, which was recommended by reviewer B78P. The evaluation shows that the flow matching (+simulator) posteriors have good coverage, albeit not perfect. They perform better than the MCMC-based baselines HMC and AIES for the given computational budget.
- **Updated plots**: We have updated the plots showing posteriors for the two systems in figure 15 and 16, which now include the posterior for flow matching without simulator feedback. Including simulator feedback visibly and consistently reduces the bias of the posterior.

Overall, we appreciate the critical feedback on the experiments. We have identified two main concerns that we would like to address:

- **Flow matching on par/not significantly improving baselines in toy SBI problems** Our main contribution of this paper was not to show that flow matching/diffusion improves existing low-dimensional benchmark tasks. There are already several recent works that advocate for diffusion training in SBI, e.g. [4,5,6]. As discussed in the paper, flow matching has been evaluated for these exact SBI tasks and compared against other neural posterior estimation (NPE) methods in [4], where it also did not outperform all baselines in every scenario. Diffusion/flow matching provides a stable training algorithm, scales to large network architectures and also works well when parameters/observations are high-dimensional - which is not the case for most NPE methods. We included table 1 with baselines comparisons for the SBI toy tasks in section 5.1, since we do follow-up experiments with flow matching in section 5.2 and 5.3. The main contribution of this paper is to introduce simulator feedback as an extension to flow matching for sbi, as we have identified that offline learning from data alone is not sufficient to obtain very accurate posteriors for many problems.
- **Gravitational lensing experiment too low-dimensional/possible bias in predicted posterior** The parameter space comprises 23 dimensions. While this is small in comparison to more high-dimensional data such as images, the marginal posterior distributions for the parameters can be very narrow, as the simulator is very sensitive to some of them. In addition to that, while the parameter space is relatively small, the observations, which are images, are high-dimensional.
Therefore this is a very difficult problem and some of the modeled systems still show visible residuals. We have added additional baselines as mentioned above and included an additional coverage test, which shows that flow matching has better coverage than the other MCMC-based methods with the given computational budget.

[1] Wu, L., Trippe, B., Naesseth, C., Blei, D., & Cunningham, J. P. (2023). Practical and asymptotically exact conditional sampling in diffusion models. Advances in Neural Information Processing Systems, 36.

[2] Song, J., Zhang, Q., Yin, H., Mardani, M., Liu, M. Y., Kautz, J., ... & Vahdat, A. (2023, July). Loss-guided diffusion models for plug-and-play controllable generation. In International Conference on Machine Learning (pp. 32483-32498). PMLR.

[3] Lemos, P., Coogan, A., Hezaveh, Y., & Perreault-Levasseur, L. (2023, July). Sampling-based accuracy testing of posterior estimators for general inference. In International Conference on Machine Learning (pp. 19256-19273). PMLR.

[4] Wildberger, J., Dax, M., Buchholz, S., Green, S., Macke, J. H., & Schölkopf, B. (2024). Flow matching for scalable simulation-based inference. Advances in Neural Information Processing Systems, 36.

[5] Sharrock, L., Simons, J., Liu, S., & Beaumont, M. (2024). Sequential neural score estimation: Likelihood-free inference with conditional score based diffusion models. In Proceedings of the 41st International Conference on Machine Learning, PMLR.

[6] Gloeckler, M., Deistler, M., Weilbach, C. D., Wood, F., & Macke, J. H. (2024). All-in-one simulation-based inference. In Proceedings of the 41st International Conference on Machine Learning, PMLR.

---

### Meta-Review · Area_Chair_pxQg · 2024-12-20

**Metareview:**

The reviewers are somewhat divided (3-3-3-6-6) about the paper, but they overall lean towards rejection. The paper introduces simulator feedback as an extension to flow matching for simulation-based inference. The approach is well-motivated, but the results are not convincing. The author-reviewer discussion has been constructive and has led to a number of clarifications and improvements, with the addition of a few new results. However, the reviewers still believe substantial improvements are needed, in particular regarding the presentation, the clarity, and the evaluation of the approach. For these reasons, I recommend rejection. I encourage the authors to address the reviewers' comments and to resubmit to a future conference.

**Additional Comments On Reviewer Discussion:**

The author-reviewer discussion has been constructive and has led to a number of clarifications and improvements, with the addition of a few new results.

---

### Decision · Program_Chairs · 2025-01-22

Reject